# Single-virus genomics reveals hidden cosmopolitan and abundant viruses

Francisco Martinez-Hernandez[1], Oscar Fornas[2,3], Monica Lluesma Gomez[1], Benjamin Bolduc[4], Maria Jose de la Cruz Peña[1], Joaquín Martínez Martínez[5], Josefa Anton[1], Josep M. Gasol[6], Riccardo Rosselli[7], Francisco Rodriguez-Valera[7], Matthew B. Sullivan[4,8], Silvia G. Acinas[6] & Manuel Martinez-Garcia[1]

Microbes drive ecosystems under constraints imposed by viruses. However, a lack of virus genome information hinders our ability to answer fundamental, biological questions concerning microbial communities. Here we apply single-virus genomics (SVGs) to assess whether portions of marine viral communities are missed by current techniques. The majority of the here-identified 44 viral single-amplified genomes (vSAGs) are more abundant in global ocean virome data sets than published metagenome-assembled viral genomes or isolates. This indicates that vSAGs likely best represent the dsDNA viral populations dominating the oceans. Species-specific recruitment patterns and virome simulation data suggest that vSAGs are highly microdiverse and that microdiversity hinders the metagenomic assembly, which could explain why their genomes have not been identified before. Altogether, SVGs enable the discovery of some of the likely most abundant and ecologically relevant marine viral species, such as vSAG 37-F6, which were overlooked by other methodologies.

[1] Department of Physiology, Genetics, and Microbiology, University of Alicante, Carretera San Vicente del Raspeig, San Vicente del Raspeig, Alicante 03690, Spain. [2] Centre for Genomic Regulation (CRG), The Barcelona Institute for Science and Technology (BIST), Carrer del Doctor Aiguader, 88, PRBB Building, Barcelona 08003, Spain. [3] Universitat Pompeu Fabra (UPF), Carrer del Doctor Aiguader, 88, PRBB Building, Barcelona 08003, Spain. [4] Department of Microbiology, The Ohio State University, 105 Biological Sciences Building, 484 West 12th Avenue Columbus, Ohio 43210, USA. [5] Bigelow Laboratory for Ocean Sciences, 60 Bigelow Drive, PO Box 380, East Boothbay, Maine 04544, USA. [6] Department of Marine Biology and Oceanography, Institut de Ciències del Mar (ICM), CSIC, Passeig Marítim, 47, Barcelona 08003, Spain. [7] Evolutionary Genomics Group, Departamento de Producción Vegetal y Microbiología, Universidad Miguel Hernández, Campus San Juan, San Juan, Alicante 03550, Spain. [8] Department of Civil, Environmental and Geodetic Engineering, The Ohio State University, The Ohio State University, 105 Biological Sciences Building, 484 West 12th Avenue Columbus, Ohio 43210, USA. Correspondence and requests for materials should be addressed to M.M.-G. (email: m.martinez@ua.es).

Viruses are the most abundant biological entities on Earth and a major reservoir of genetic diversity[1] that hide an enormous complexity across all habitats[2–5]. Despite the role of viruses in shaping microbial ecosystems[1,4–6], global diversity patterns of viral communities[3,4] are only beginning to be elucidated for certain environments[2,3,5,7,8]. Culture-based methods inefficiently capture naturally occurring viral diversity[1]. As a consequence of the inability to cultivate the majority of microbial hosts, most bacterial and archaeal phyla lack known viruses[4,9]. In turn, culture-independent approaches have provided a wealth of genetic information on environmental viral communities. Metagenomic studies have delivered thousands of viral genomes and large genome fragments. For instance, metagenomic strategies based on capturing viral genomes in fosmids have broaden our knowledge on abundant and widespread viruses in surface waters[10] and in the deep Mediterranean Sea[11]. Broader metagenomic surveys in the context of the *Tara* Oceans expeditions have unveiled ocean viral community patterns at a global scale[3] and provided a map of abundant, double-stranded DNA viruses with a total of 15,222 epipelagic and mesopelagic viral populations, comprising 867 major viral clusters, each corresponding to approximately genus-level groupings[2]. Such studies emphasize the large disparity with cultivation efforts, as <1% of the observed viral populations are represented in culture[2,3,10,11]. However, even with these greatly augmented reference databases, available reference genomes—cultivated and uncultivated—fail to recruit most (>80%) viral metagenomic reads[2]. Thus, there is an agreement that much viral diversity remains to be discovered in the oceans.

Over the last years, single-cell genomics (SCGs) has enabled sequencing of individual genomes of many abundant and ecologically important prokaryotes in marine and other environments[12–16] by disentangling the genetic complexity of the community to the minimum level, the cell. This powerful approach has opened up new frontiers that overcome some of the metagenomic assembly limitations and culture biases. SCGs also provides the means for a better understanding of the biology, ecology and evolution of microbial communities[14,15]. Currently, a major bottleneck in metagenomics is the reconstruction of genomes from closely related strains. Furthermore, metagenomic assembly 'obscures' the population microdiversity by delivering consensus genome contigs that hide the extant genetic heterogeneity. In many cases such information is crucial for a comprehensive understanding of virus–host interactions and dynamics[17]. Although metagenomics binning of assembled contigs into species clusters has been a major advancement in metagenomics[18], binning at the strain level remains a technical challenge. Albeit not exempt of biases[16], SCGs simplifies the complexity of the puzzle, by assembling individual genomes, one at a time, and therefore captures the natural genetic variability[15]. The feasibility of adapting SCGs methodology to virology has been demonstrated by two previous studies[19,20], yet neither addressed the issue at the level of single viruses in natural viral assemblages. One study sorted and sequenced individual viral particles from a bacteriophage culture of lambda and T4 of *Escherichia coli*[19], and the second study employed fluorescence-activated virus sorting (FAVS) and whole-genome amplification (WGA) to recover the genetic information of a pool of 5,000 sorted uncultured viruses from a marine sample[20]. Oceans have been extensively studied by viral metagenomics and culturing, and thus represent a model scenario to test whether portions of marine viral communities are missed by these techniques. We hypothesize that high intra-population viral diversity could lead to ambiguous sequence metagenomic reconstruction and/or hinder the genome assembly of abundant uncultured viruses.

Here we employ single-virus genomics (SVGs) to natural marine viral assemblages from the Mediterranean Sea (epi- and mesopelagic) and the deep Atlantic Ocean, and demonstrate the power of this approach to uncover the genomics of some of the most abundant marine viruses.

## Results

**SVGs of marine viruses.** First, using FAVS in combination with confocal fluorescence microscopy, we demonstrated the suitability of the used flow cytometer sorter to separate individual viral particles from a culture isolate (Supplementary Fig. 1 and Supplementary Notes 1 and 2). Subsequently, a total of 2,234 virus-like particles were sorted by FAVS from environmental seawater samples collected from the Atlantic Ocean during the Malaspina expedition (bathypelagic, 4,000 m depth) and from the Mediterranean Sea (surface and deep chlorophyll maximum, 60 m depth) (Table 1; Supplementary Figs 2–4). WGA of the sorted single viral particles yielded a total of 392 marine viral single-amplified genomes (vSAGs) (Table 1; Supplementary Fig. 4 and Supplementary Note 1). Forty-four of these vSAGs were selected at random for Illumina sequencing (Table 1; Supplementary Table 1 and Supplementary Figs 5 and 6). For most vSAGs (32 out of 44), a single large genome contig was obtained (from ≈10–78 kb, mean ≈20 kb, Supplementary Table 1 and Supplementary Note 3). Genome annotation[21] and protein-sharing network[2] analysis confirmed that vSAGs (Fig. 1) were viruses and no other types of biological particles, such as marine vesicles or gene transfer agents[22]. Most of the vSAGs (n = 22) were tentatively assigned to the Caudovirales (Fig. 1; Supplementary Table 2 and Supplementary Note 3) representing putative novel viral species (n = 37), and genera (n = 7) from cosmopolitan oceanic virus clusters (VCs)[2] (Fig. 1; Supplementary Fig. 7 and Supplementary Tables 2–4). Compared to viral genome fragments assembled from the recent Global Oceanic Virome (GOV) metagenomics data set[2], our 37 vSAGs representing putative new viral species were at the 'core' of VCs in the global marine viral network (Fig. 1; Supplementary Fig. 7 and Supplementary Tables 2–4). The centrality of vSAGs within VCs indicated higher frequency of shared proteins with other uncultured viruses (∼11 shared proteins per vSAGs with ∼60% of amino-acid identity). However, the remaining seven

**Table 1 | Summary of marine samples and viral single-amplified genomes (vSAGs).**

| Sample | Treatment* | No. of sorted single viruses | No. of vSAGs | No. of sequenced vSAGs |
|---|---|---|---|---|
| Mediterranean Sea (Barcelona Beach) | A | 332 | 63 | 8 |
| Mediterranean Sea (Blanes Bay Microbial Observatory) | B | 664 | 149 | 21 |
| | C | 332 | 41 | 11 |
| Mediterranean Sea (DCM) | B | 332 | 76 | 1 |
| North Atlantic Ocean | E | 664 | 63 | 3 |
| Total | | 2,324 | 392 | 44 |

*Treatments: A, fixed sample + liquid $N_2$ and KOH (pH 14) shock; B, unfixed sample + liquid $N_2$ and KOH (pH 14) shock; C, unfixed sample + KOH (pH 14) shock; E, cryopreserved in GlyTE + treatment.

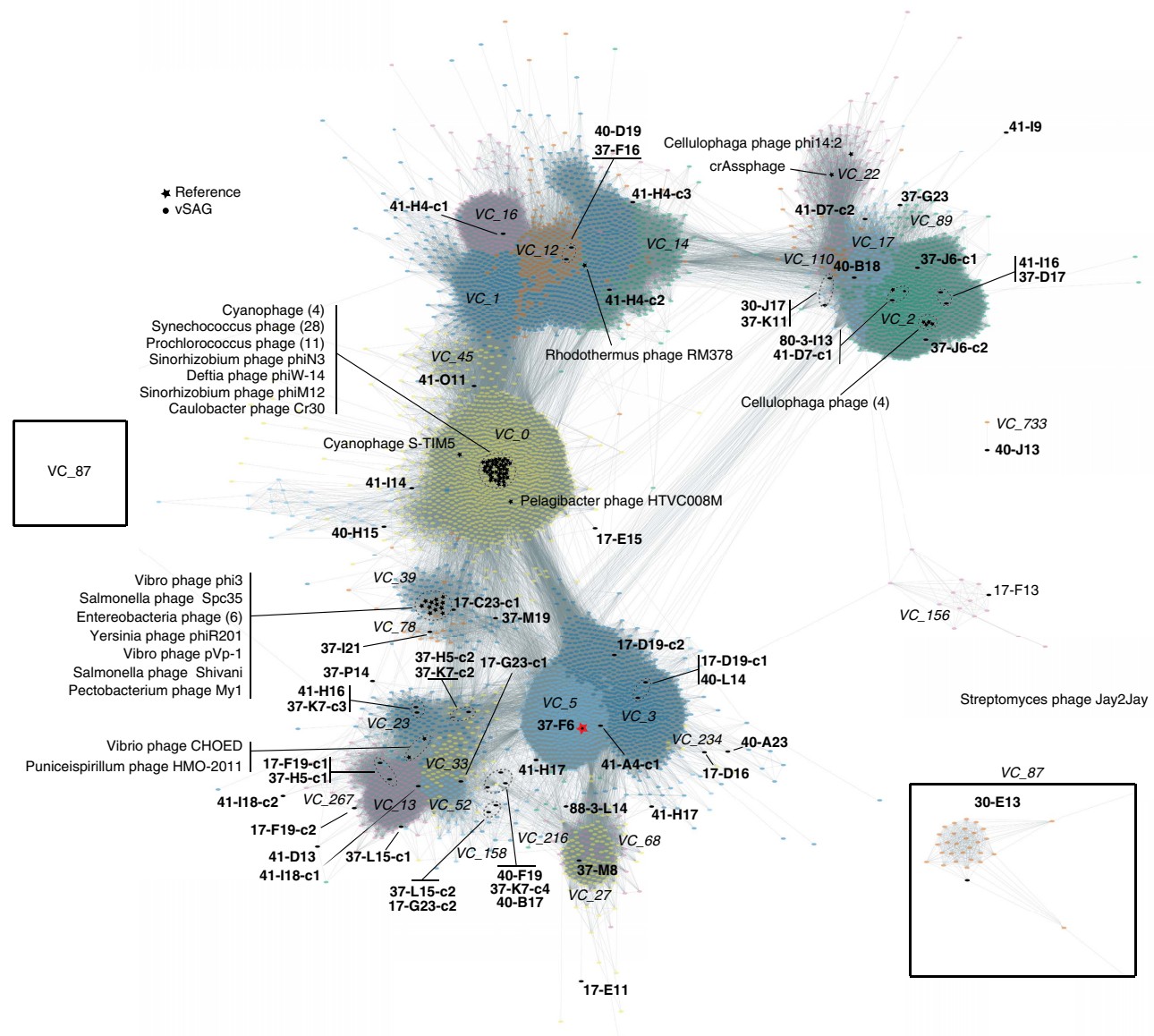

**Figure 1 | Global viral protein-sharing network.** A total of 5,539 partial and full-length genomes, and 634,497 relationships (edges) from GOV[2], environmental phage from Genbank, archaeal and bacterial viral references (indicated by a black star, *), and vSAGs (this study, indicated by a black dot ●, bold font) were included in the analyses. Only viral clusters—with each viral cluster indicated by unique colours—including ≥1 vSAG sequences are represented. Edges between nodes indicate a statistically significant weighted pairwise similarity between the protein profiles of each node (see Methods) with similarity scores ≥1. Viral clusters (italic font) are determined by applying the Markov Cluster Algorithm (MCL) to the edges[2]. vSAG 37-F6 is indicated by a red star.

vSAGs (Supplementary Table 4), such as the autochthonous virus 88-3-L14 from the unexplored massive bathypelagic Atlantic Ocean biome (4,000 m depth) (Fig. 1; Supplementary Figs 7 and 8), had weaker connections within the viral network and were placed on the periphery of VCs, (Supplementary Fig. 7); consequently, they could represent novel genera. Only the surface vSAGs 41-O11, 37-K7 and 37-L15 were distantly related to known phage isolates (cyanophages and Pelagibacter phages; genome average nucleotide identity (gANI) <50%; Supplementary Fig. 9).

**Global abundance of single viruses**. To assess the global abundance of our vSAGs relative to other approaches (viral metagenomics or viromics and culturing), we analysed fragment recruitment based

abundances in viromic[2,3,10,23,24] and microbial metagenomic[25] data sets. For virome recruitment, ≥95% and ≥70% nucleotide identity thresholds were used to target reads from viruses identical to or within the same species[10] than our vSAGs and from the same genus or subfamily viral taxa, respectively[2] (Supplementary Fig. 10; Supplementary Notes 4 and 5). Overall, comparative recruitment of viral genomes obtained by different approaches against available viromes indicated that in all cases but one (Northwest Arabian Sea upwelling virome) the recruitment mean was higher (ANOVA *P* value <0.001) for our vSAGs followed by other methods as follows: SVGs > virus cloned in fosmids[10] > viruses from single bacterial cells[26] > virome contigs[2,3] (*Tara* Oceans Viromes (TOV)) > virus isolates (Fig. 2; Supplementary Figs 11–13 and Supplementary Note 4). Furthermore, this analysis indicated that our surface vSAGs were highly abundant both at the sampling sites as

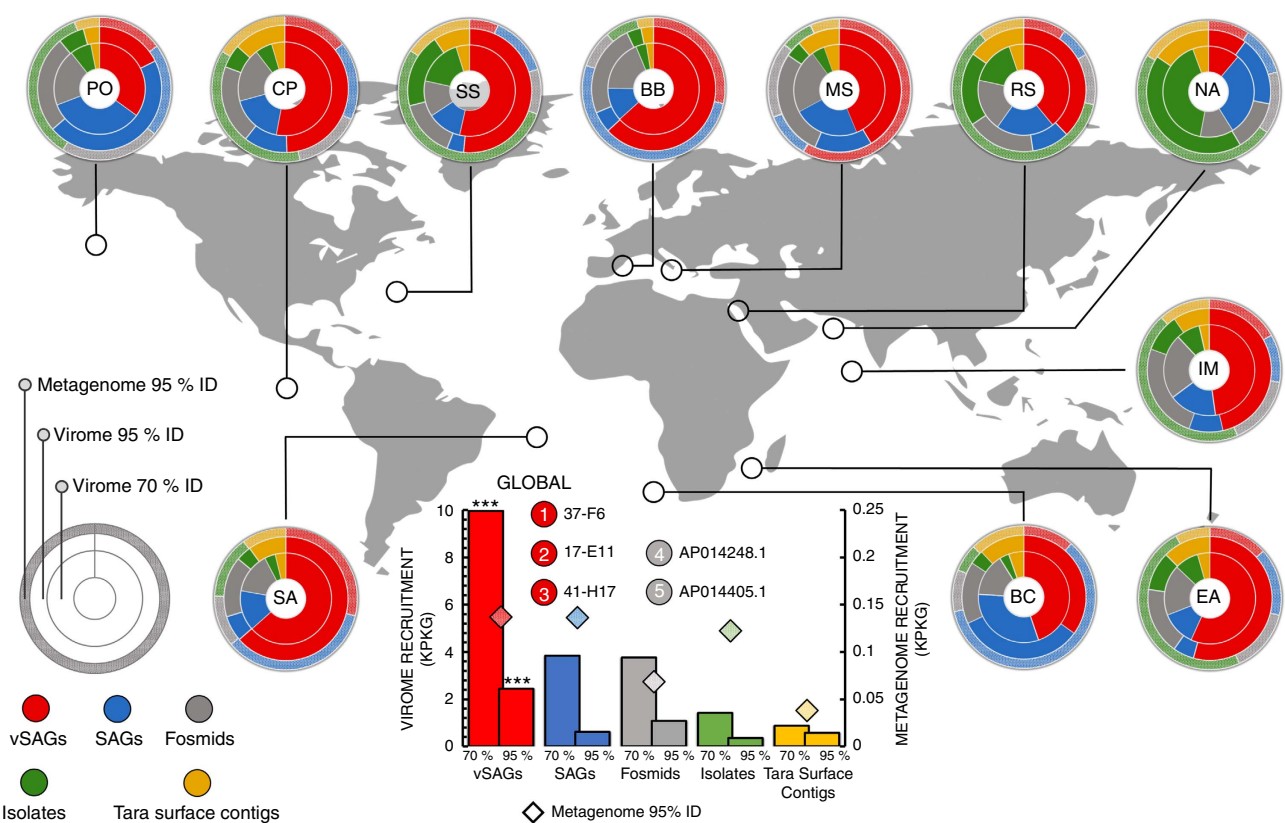

**Figure 2 | Relative abundance and distribution of surface marine viruses.** Virome and microbiome metagenomic fragment recruitments of marine viruses in each ocean. Rings represent the relative microbiome and virome recruitment frequency for each genomic data set, corresponding to the relative abundance of viral populations. External ring is for the microbiome recruit by using ≥95% nucleotide identity threshold (species level). Inner and medium rings depict the virome recruitment at two different nucleotide identity cut-offs, ≥70% and ≥95%, corresponding to the genus and species levels, respectively (Supplementary Fig. 10 and Supplementary Notes 4 and 5). Viral genomic data sets used were: 40 surface vSAGs (this study), 179 reference virus isolates (Supplementary Table 9), 1,148 viral fosmids[10], 20 viral genomes from uncultured prokaryotic single cells[26] and 3,018 surface viral contigs from the *Tara* expedition[3]. For this calculation: (1) normalized recruitment as the total recruited nucleotides (kb) per kb of viral genome per Gb of virome (KPKG) was estimated for each virus genome, (2) mean normalized recruitment was calculated for each virus genomic data set (see also Supplementary Fig. 11) and (3) mean was normalized by the sum of means from all virus genomic data set expressed as relative recruitment. Statistically significant differences between the recruitment frequency average of the vSAGs versus the rest of viral groups are indicated (***; ANOVA *P* value <0.001). Viromes and microbiomes from previous surveys[3,23–25] used here are abbreviated as: Pacific Ocean (PO), Chile-Peru oceanic region (CP), South Atlantic (SS), Red Sea (RS), Mediterranean Sea (MS), Northwest Arabian Sea upwelling (NA), Indian Monsoon gyre province (IM), Eastern Africa Coastal Province (EA), Benguela Current (BC) and Sargasso Sea (SS). The microbiome and virome from Blanes Bay Microbial Observatory (BB), where surface vSAGs were obtained, was constructed in this study. Global oceanic viromics and microbiome fragment recruitments for each virus genomic data set is represented in the centre of the picture, circles represent overall oceanic top-5 ranking of most abundant viruses at the species level (≥95% of identity).

well as throughout many different surface oceanic regions (Fig. 2; Supplementary Figs 11, 13 and 14; Supplementary Table 5 and Supplementary Note 4). Furthermore, the recovered vSAGs showed the highest relative microbiome recruitment frequency against the *Tara* Oceans microbiome data sets (Fig. 2). Microbiomes are known to contain significant amounts of viral DNA derived from cells undergoing the lytic cycle, and this has been commonly used to determine abundances and diversity of marine viruses in cellular metagenomic libraries[10]. Though it might also be possible that some free virus particles had been retained onto the filters, our microbiome recruitment suggest that those viruses may have been actively infecting the marine bacterioplankton (0.2–3 μm size).

The quantitative analysis of the vSAGs abundance indicated that the virus vSAG 37-F6 along with 17-E11 (distantly related to virus 37-F6; ≈60% gANI) and 41-H17 were more abundant, at the species level, in the global marine surface viriosphere than any extant dsDNA virus in public data sets (Figs 2 and 3; Supplementary Fig. 15), including novel uncultured viruses recently described within cosmopolitan VCs from the GOV data set[2]. For

the remaining marine virome data sets, the most abundant viruses for each category were the viruses AAA164-I21 and AAA160-P02, found in two uncultured sorted single cells belonging to Verrucomicrobia and Flavobacteria[26], respectively, a putative cyanophage cloned in the fosmid AP014248.1 (ref. 10), the Pelagibacter phage strain HTVC010P (ref. 27) and the *Tara* contig 34DCM_32712 (ref. 3) (Fig. 3; Supplementary Fig. 15). Estimated abundance of RNA and ssDNA viruses[28] was not considered in this study. In the deep ocean, the vSAG 88-3-L14 from the Atlantic Ocean (4,000 m) representing a potentially novel genus was also fairly cosmopolitan and abundant across distant bathypelagic habitats (Supplementary Fig. 16), with no viral relatives in public databases (Fig. 1; Supplementary Tables 2 and 4).

**vSAG 37-F6 is the putative most abundant marine virus.** The biogeographic analyses of the vSAG 37-F6 indicated that, at the genus and species level, it was highly abundant in several oceanic regions, such as the Atlantic and Indian oceans; even more

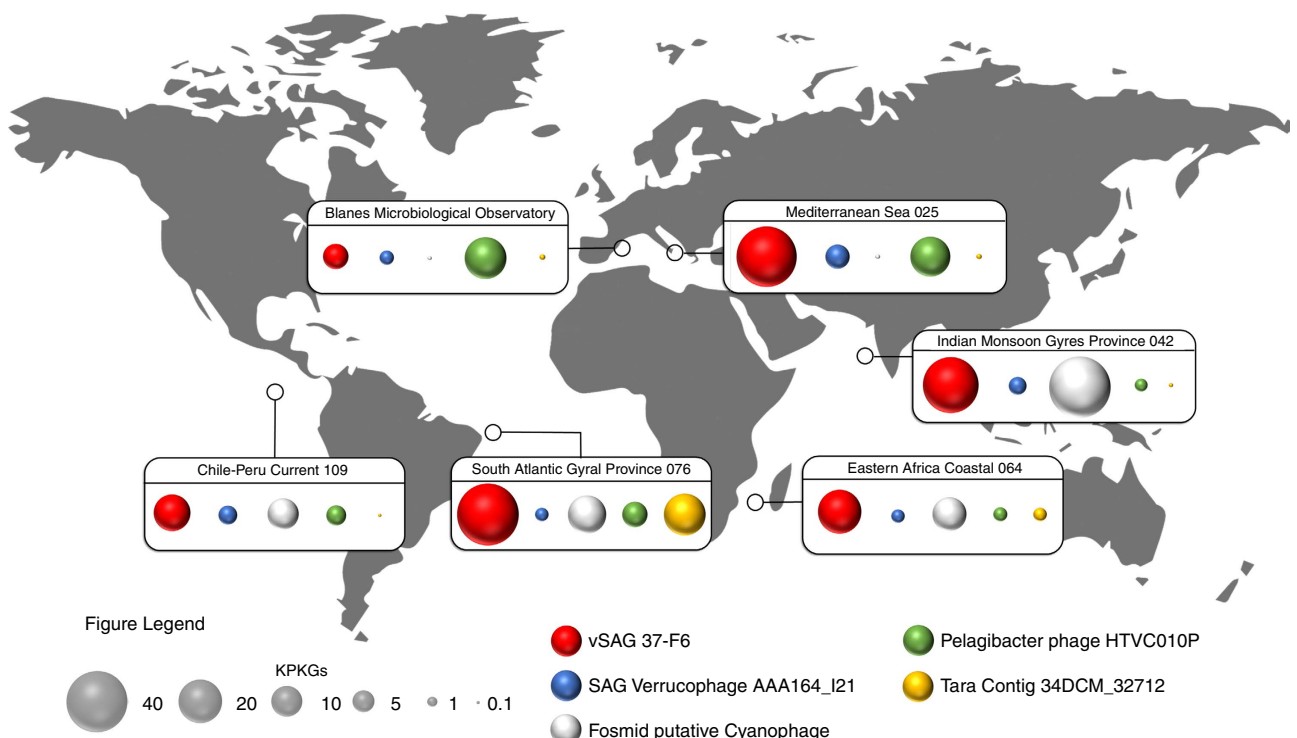

**Figure 3 | Biogeography of most abundant marine viruses.** The abundance of the most abundant surface dsDNA viruses for each virus genome data set according to the procedure for genome recovering (single-virus genomics (red), viruses from single bacterial cells[26] (blue), virus cloned in fosmids[10] (grey), virus isolates (green) and viromics from *Tara* Oceans data set (yellow)[2,3]. Fragment recruitment data were used to estimate the overall abundance for each region. Bubbles represent the fragment recruitment estimation expressed in KPKG (as in Fig. 2).

abundant than at the sampling point (Blanes Bay Microbial Observatory; Mediterranean Sea) (Figs 2 and 3; Supplementary Fig. 15). When comparing to well-known predominant viral species isolates, such as the Pelagibacter phage HTVC010P28, the vSAG 37-F6 was the most abundant virus in 20 of the 24 metaviromes analysed (Fig. 3; Supplementary Fig. 15). A detailed genomic comparison of the vSAG 37-F6 revealed that it was not genetically similar to any known virus isolates (Fig. 4; Supplementary Figs 10a and 14). Only three uncultured viruses showed genome synteny and relatively low genetic relatedness (<63% nucleotide identity) against vSAG 37-F6: the above-mentioned verrucophage (AAA164-I21) and flavophage (AAA160-P02) from single cells[26,29], and a third virus cloned in a fosmid from a deep Mediterranean Sea water sample (3,000 m depth)[11] (Fig. 4; Supplementary Fig. 17 and Supplementary Table 6). Finding a distant vSAG 37-F6 viral relative at such depths (Fig. 4b) along with a significant fragment recruitment of that virus in the deep ocean virome data sets[11] (Supplementary Fig. 15) suggests that 37-F6-like viruses likely populate the deep ocean as well. We were not able to identify the putative host for the virus 37-F6 based on *in silico* host prediction by using *k-mer* analysis[30], identification of CRISPR host spacers[2], or tRNA signatures (Supplementary Fig. 17).

**Mining viral signals of vSAGs in proteomic data.** Recently, the most abundant viral marine proteins detected by proteomics in different oceans were identified as capsid proteins of predominantly unknown marine viruses of the cluster CAM_CRCL_773 (ref. 31). The capsid protein encoded by gene 9 of vSAGs 37-F6 was homologous to these unknown abundant capsid proteins (Fig. 5a). The closest capsid protein was that of the phage AAA160-P02 (86% amino-acid identity) recovered in a flavobacterium single cell

(Fig. 5b). The predicted three-dimensional (3D) structure of vSAG 37-F6′s gene 9 was nearly identical to the 3D model previously proposed for the capsid proteins of the cluster CAM_CRCL_773 (Fig. 5b)[31].

Furthermore, when comparing the predicted peptide sequences ($n = 4,871$) obtained by mass spectrometry (MS) from the South Atlantic and Indian Oceans and the Mediterranean Sea viral proteomes[31] against the *in silico* digested capsid protein sequences of virus 37-F6, over 200 peptides from all oceanic regions were a perfect match (100% identity) (Figs 4a and 5). When the comparison of peptide sequences from MS data was extended to the whole viral genome data sets, results showed that predicted capsid proteins of vSAGs accumulated a high number of recruited peptides (Fig. 5; Supplementary Fig. 18). In particular, the capsid proteins of vSAG 37-F6 along with 41-A4, and phage AAA160-P02 showed the highest rate of proteomic recruitment in the *Tara* viral proteome data set. In addition, these capsid proteins of vSAGs were also abundant in a bacterioplankton proteome from the Oregon Coast (Supplementary Fig. 19; Supplementary Table 7). Thus, metagenomic and proteomic data point to the ubiquity and high abundance of some of the uncultured viral species recovered by SVGs.

**Microdiversity affects metagenomic assembly.** To explain why our discovered abundant viruses have been overlooked by metagenomic assembly[2,3,23], we take advantage of an intriguing empirical observation from the species/genus-specific recruitment pattern of viral populations (hereafter as diversity curve) obtained in the different viromes. In our study, the abundant surface vSAGs populations showed high accumulated microdiversity in the diversity curves against the viromes from the corresponding sampling sites (Fig. 6a; Supplementary Fig. 14 and Supplementary

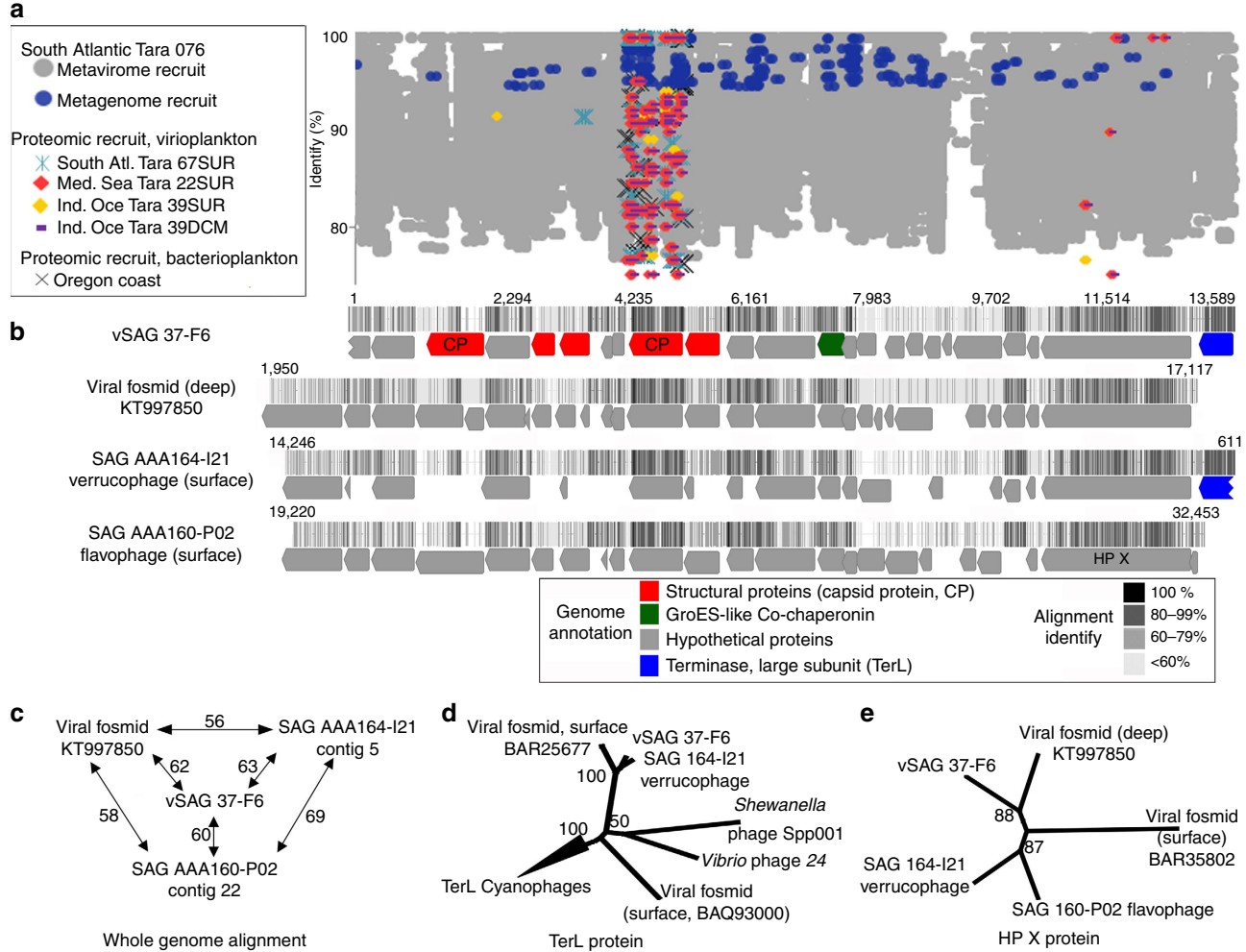

**Figure 4 | Ecogenomics of the putative most abundant surface marine virus, the vSAG 37-F6.** (**a**) Virome, microbial metagenome and proteome fragment recruitment in different data sets[3,31,62]. A hypervariable genomic island in virus 37-F6 was detected between genomic position 9,000 and 9,700 (unknown protein). (**b**) Genome annotation, synteny and whole-genome alignment of vSAG 37-F6 with closest viral relatives[26,29]. Colour in the alignment (from black to white) denotes identity values among all four genomes for each genome position. (**c**) Whole-genome similarity with closest viral relatives. (**d,e**) Phylogeny of large subunit of viral terminases (TerL) and the large conserved hypothetical protein X (HP X) based on maximum-likelihood method. Bootstrap values are indicated in nodes.

Note 5). However, in comparison to viromics for those abundant viral population from *Tara* data set[3], a contrasting population structure lacking microdiversity was observed (Fig. 6a). Furthermore, the frequency of single-nucleotide polymorphisms (SNPs) in the vSAG 37-F6 species population was ∼10 to 250-fold higher, depending on the geographical origin of the sample, than in the most abundant viruses recovered by viromics[2,3] (Fig. 6b). Therefore, we hypothesized that population microdiversity would hinder genome reconstruction by metagenomic assembly, which may explain why metagenomics have so far failed to recover some very abundant marine viruses, as is the case for those identified in this study employing SVGs. We tested this hypothesis by creating simulated viromics data sets with variable levels of microdiversity (Supplementary Fig. 20) that we then analysed following standard metagenomics assembly tools. We simulated three different scenarios and introduced viral populations of the vSAG 37-F6 with different degrees of microdiversity (no diversity, low/medium and high) within natural *Tara* viromes from the Mediterranean Sea (for details, see Methods, Supplementary Fig. 20 and Supplementary Note 5). Our results showed that common metagenomic assembly strategies delivered complete reference viral genomes from

simulated data sets in the absence of population microdiversity, or when it was very low (Fig. 6c; Supplementary Fig. 20 and Supplementary Note 5).

## Discussion

Detection of viral particles by flow cytometry is highly dependent on optimum fluorescence staining of the viral nucleic acids as well as on the equipment sensitivity, since scattered light and fluorescence signals are close to the detection limit of the instrument. Sorting of single viruses at very low flow rate was critical to prevent coincident events of multiple viral particles. In our case, the estimated ratio of putative sorted particle versus generated drops (see Methods for details) ensured sufficient separation between each sorted particle to prevent sorting of doublets (two viruses in the same droplet), which could later obscure the interpretation of SVGs data. The standard protocol commonly used for staining and detecting viruses by flow cytometry employs high concentration of a fixative agent (0.5% glutaraldehyde)[32], which ultimately would prevent the amplification of genetic material. Here the standard procedures for staining viruses were adapted and optimized without

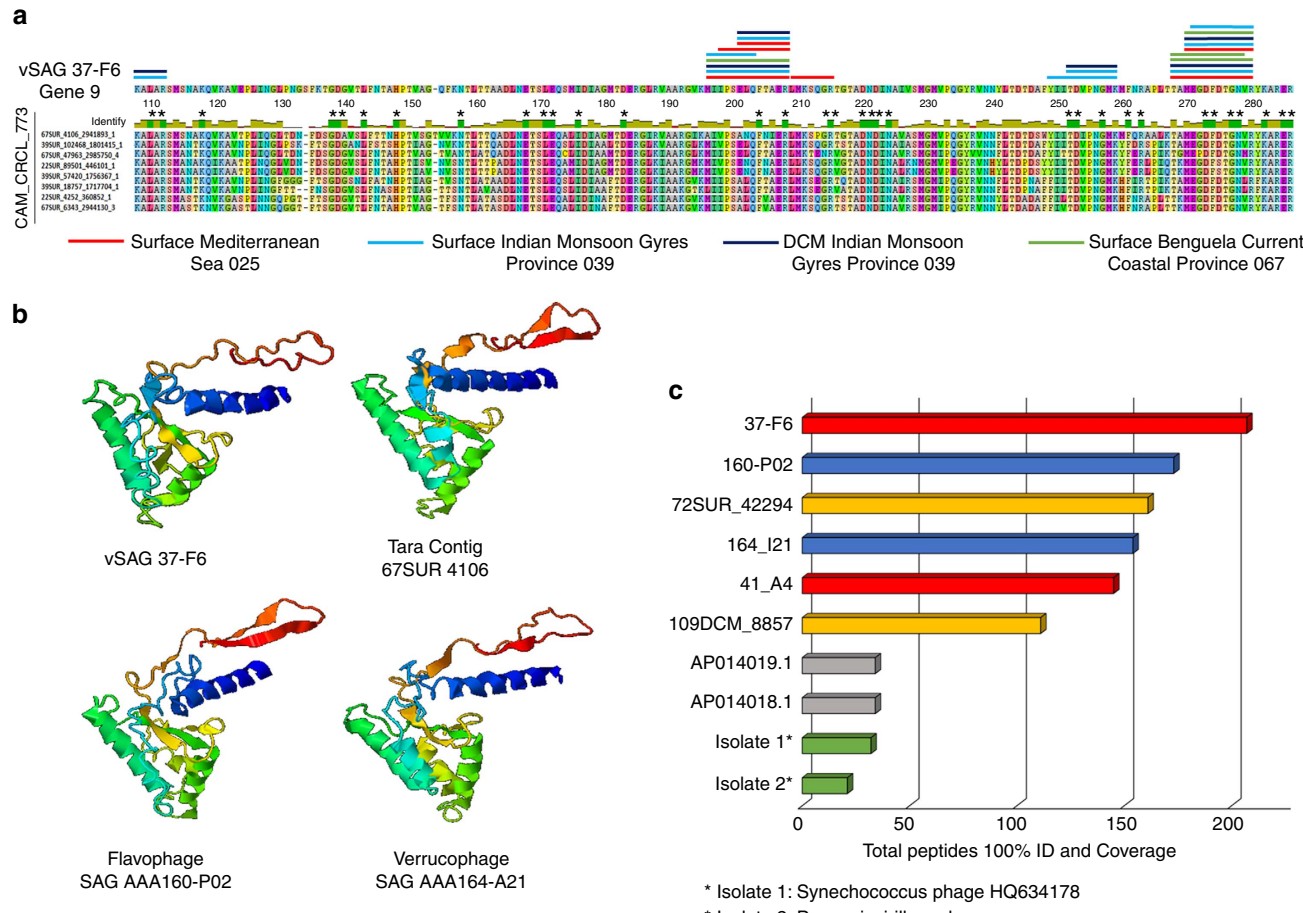

**Figure 5 | Capsid protein of vSAG 37-F6 and abundance in proteomic *Tara* viral data set.** (**a**) Peptide alignment of vSAG 37-F6 with the capsid proteins of cluster CAM_CRCL_773. For convenience, we only show eight protein sequences out of 152 total capsid proteins. Coloured lines above amino-acid sequence of vSAG 37F6 represent the perfect matches of predicted peptide sequences from *Tara* expedition[31] (100% identity similarity and query coverage). Colour denotes the origin of peptides. Conserved amino-acid positions in the protein alignment are denoted with '*' (**b**) Representative 3D-structural model, using I-TASSER prediction server, of the 37-F6 capsid protein compared with the nearest viral capsid proteins: the *Tara* Contig 67SUR_4106 and viruses from SAGs AAA160-P02 (*Flavobacteria*) and AAA164-I21 (*Verrucomicrobia*). (**c**) Number of total recruited peptides from *Tara* expedition[31] (100% identity and query coverage) for the top two most recruiting viruses from each viral genomic data set[3,10,26].

apparently missing major viral populations (Supplementary Fig. 2). During FAVS, as with SCGs[29], viral stained particles are sorted at random, which means that the more abundant a virus is within a sample, the higher its probability to be sorted. Thus, the uncultured viruses sorted in this study, represent a random subset of, likely, the most abundant dsDNA virus members within natural viral assemblages. Extraction of viral DNA from the capsids of single-sorted viruses without degrading genetic material is critical for the success of WGA. As with SCGs, the proper breakdown of the capsid is paramount to guarantee the success in downstream analyses. A combination of KOH buffer and liquid nitrogen shock proved to be efficient for lysing the viral capsids from marine samples (Supplementary Fig. 4). Although our protocol is promising for a wide range of capsid types, further experiments will need to be conducted to assess the general feasibility of the method. We are aware of the possibility that sub-optimal lysis of some virus groups (either degrading or not releasing the nucleic acids) might have led to underrepresentation of certain virus groups.

Metagenomic fragment recruitment has been widely used to assess the abundance of marine viruses[3,10,11,27] and several programs are available to perform fragment recruitment, such as BLAST[10,11,27] or Bowtie[3]. Here we tested the impact of different recruitment algorithms on our results, and in particular, the reciprocal best-hit approach (each query read assigned only to one viral genome by best-hit score). Our results (Supplementary Fig. 21) indicated no significant differences among the different recruitment strategies. Thus, our data confirmed the overwhelming high relative recruitment rate of our vSAGs and suggest that several of the viruses reported here, in particular vSAG 37-F6, are putatively the most widely distributed, abundant, and likely active virus at the genus- and species-level taxa identified so far in the surface viriosphere (Figs 2 and 3; Supplementary Figs 11–19).

Application of viral taxonomy criteria, such as demarcation of viral genera or species, to uncultured viruses is controversial. The International Committee of Taxonomy of Virus has recently stated[33] that taxonomy is moving to genome-based criteria in the era of metagenomics, but these criteria are currently under debate[2,34,35]. We aimed at targeting and recruiting uncultured viral populations at the species (very closely related) and genus (or subfamily) taxonomic levels by carrying out virome recruitment at 95% (refs 2,10) and 70% cut-off levels, respectively. In our study, nearly all obtained single viruses represented potentially new viral species, and in some cases likely new genera as well, that are highly abundant in nature. It is worth noting that our diversity curves (Fig. 6a) indicate that some of the most abundant and

cosmopolitan vSAGs, such as 37-F6, represent viral populations with an unprecedented diversity and microdiversity at the species and genus level. Although speculative, according to predicted host ranges in the recent GOV data set[2] and taxonomical affiliation of hosts of nearest viruses to our virus 37-F6, we hypothesize that vSAG 37-F6-like populations could infect a broad range of hosts

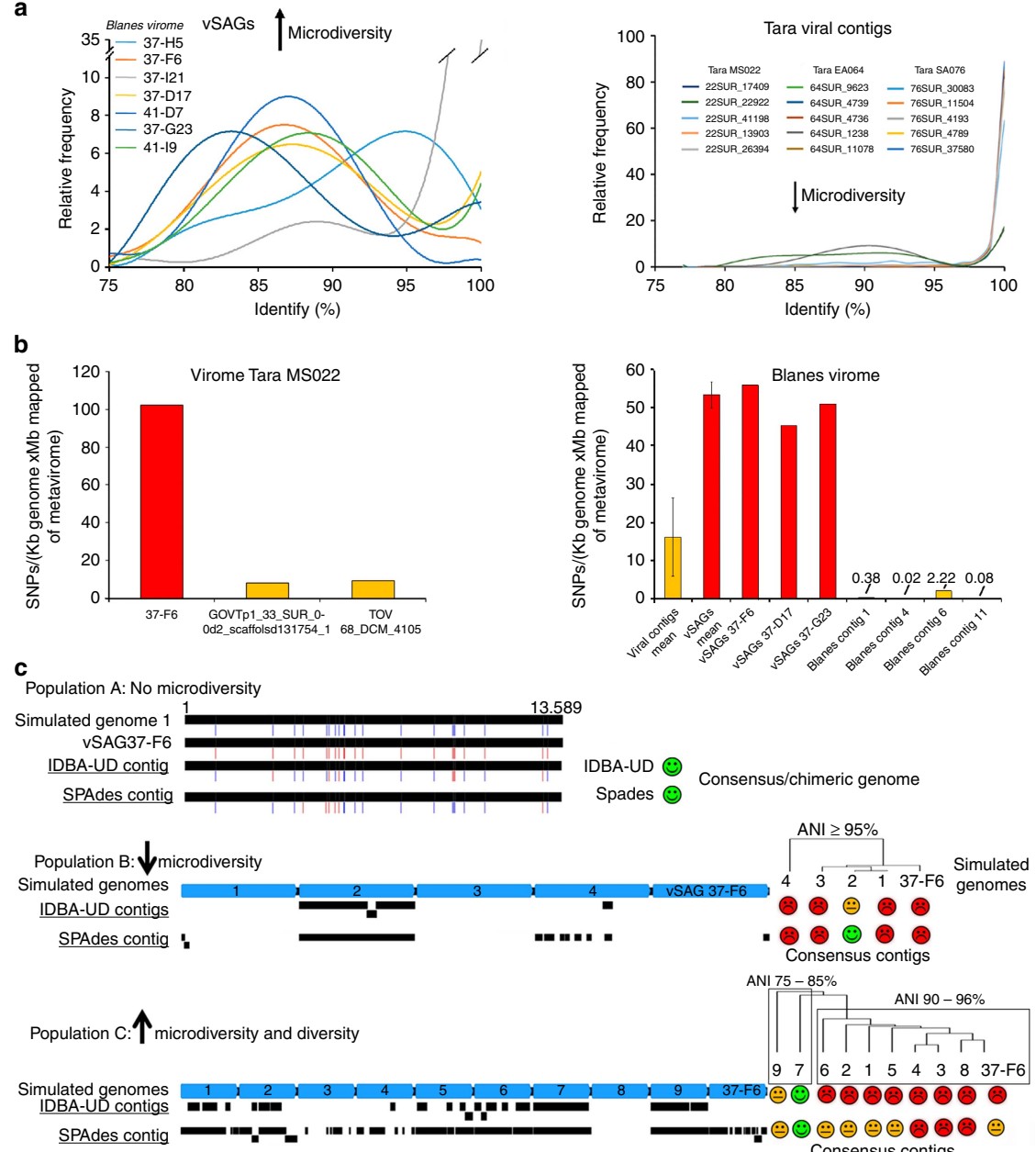

**Figure 6 | Assessment of natural vSAGs microdiversity and impact on metagenomic assembly.** (**a**) Species-specific recruitment patterns (also referred as diversity curves) for vSAGs and highly abundant viral contigs from viromics. Curves represent the percentage of recruited reads (*Y* axis) at different nucleotide identity values (*X* axis) for vSAGs and *Tara* Oceans contigs[3] in their own viromes. The five most recruiting viruses of each viral data set are shown for convenience. (**b**) SNP frequency for most abundant viral populations at the species level ($\geq$95% nucleotide identity) of vSAGs and viral contigs (within the top 30 ranking in recruitment) recovered by viromics from the Blanes Bay Microbial Observatory (same sampling site of surface vSAGs) and the *Tara* Mediterranean MS022 data set[3]. In Blanes Bay Microbial Observatory, mean ± s.d. of most abundant viral contigs (25 contigs) and vSAGs (4 contigs) are shown. (**c**) Impact of viral diversity and microdiversity on genome reconstruction by metagenomics. Three populations of virus 37-F6 with different (micro)-diversities were simulated within the virome *Tara* MS022 (ref. 3) (see details in Supplementary Fig. 20 and Supplementary Note 5). Population A lacked microdiversity (two simulated nearly identical genomes of 37-F6 with 20 SNPs). A chimeric contig with a mixture of SNPs was obtained (SNPs in blue from simulated genome 1, and in red from vSAG 37-F6). Population B simulated a simplistic scenario with five genomes (ANI$\geq$95%) without high genetic variability in the hypervariable genomic island (Fig. 4; Supplementary Fig. 14). SPAdes assembler reconstructed a consensus contig from only one of the simulated genomes. Population C simulated a more realistic microdiverse scenario than observed in panel A with 10 simulated co-existing viruses (ANI 75-95% and high variability in the genomic island (see details in Supplementary Fig. 20 and Supplementary Note 5). The genome was almost entirely assembled only from those distantly related viruses 7 and 9, while 37-F6 genome could not be assembled. Blue arrows depict the simulated genomes. Black blocks depict the resulting assembled contigs by IDBA_UD and SPAdes assemblers.

from different phyla (Fig. 4), which might partly explain the large genetic diversity within that viral population.

Several models have been proposed to unravel the 'virus–host swinging party' explaining the long-term co-existence of closely related and microdiverse virus and host strains in nature[17,36,37], and metagenomics is a common tool to address these questions[4]. In this study, we show that common metagenomic assembly strategies struggled to reconstruct viral genomes from simulated high-microdiverse populations. Furthermore, when we simulated no microdiversity with only two viral genomes with 20 SNPs of difference along the genome, the metagenomic assemblers unsuccessfully delivered a chimeric contig with a mixture of SNPs from both genomes (Fig. 6c). Accurately determining genetic viral microdiversity is crucial for gaining an under-standing of the structure and evolution of microbial population genomics[36]. For instance, a SNP within a viral species population can severely impact on viral fitness, increasing the adhesion to the host and leading to major changes in infection dynamics[38]. Our finding is in agreement with a previous study using simulated viromes from 300 virus isolates that led to similar conclusions[39]. In our study, we demonstrated the impact of microdiversity in a more realistic scenario with natural viromes and considering cosmopolitan and naturally microdiverse viral populations, such as 37-F6-like viruses. Similarly in prokaryotes, the inherent genomic complexity of many microbial populations, such as SAR11, often obfuscates facile generation of whole-genome assemblies from metagenomic data. Our data underlines the power of SVGs to tackle the genetic diversity of the uncultured viruses regardless of the existing microdiversity. We propose a 'marriage of convenience' between single-cell and metagenomics strategies, as it has been recently proposed for prokaryotes[40], to further improve the assembly of (more) complete environmental viral genomes.

Culture-based approaches are inefficient at recovering the uncultured viral majority[1,3]. In turn, shot-gun virome sequencing[2,3] is identified as the preferable tool in viral ecology. However, as we demonstrated here, virome assembly remains complicated and in many cases it yields chimeric contigs, which hide natural microdiversity (Fig. 6). Alternatively, cloning of viral genomes in fosmids has been successfully used for obtaining complete marine viral genomes[10,11], but this approach is limited by the maximum insert size that is allowed by the cloning vectors (genomes < 40 kb). Furthermore, current standard virome protocols often exclude large viruses, RNA[28] and ssDNA viruses. SVGs are already able to target ds- and ssDNA viruses. An additional benefit of SVGs is the low sample volume require-ment (typically ≤ 1 ml) to unveil the genomics of biologically relevant viruses, which is particularly advantageous for the investigation of the viral community in environments where it is technologically difficult or unfeasible, to collect large sample volumes. On the other hand, one could argue that the small sample volume may not capture the breath of a viral community (for example, due to patchy distribution). However, our results and those of a prior study using a similar approach to investigate the dsDNA viral community by bulk sorting from a single 1 ml seawater sample are comparable in diversity with viromic studies with large sample volumes[20]. Nonetheless, certain steps of the SVGs pipeline remain a challenge. For instance, the detection of viral particles with very small genomes, in particular those with ssDNA and RNA genomes, is difficult due to the low levels of fluorescence signal per viral particle achieved with commercially available fluorescence dyes. Additional stumbling blocks include complete prevention/ elimination of minute amounts of contaminant DNA, which may be amplified by WGA, as well as current biases inherent to the available WGA methods[16]. Furthermore, as it is the case

with viromics, linking individual viruses to their hosts remain a major challenge.

Altogether, our results provide evidence of SVGs enormous potential, albeit in its incipient development, to aid in unveiling the true extent of viral genetic diversity and microdiversity within natural populations and for complementing current culture and metagenomic methods for addressing key questions in environmental viral ecology. Data from this study support that vSAGs best represent uncultured dsDNA viruses in nature. Nevertheless, though not completely free of bias, recent advances on microfluidics and single-cell genome-sequencing methods[16] bode a promising road for SVGs to fill existing gaps between viromics and culture in virology.

## Methods

**Culture of bacteriophage P1.** The bacteriophage P1 of *Escherichia coli* strain LB21 (provided by Francisco Juan Martínez Mojica, Molecular Microbiology Laboratory, University of Alicante) was used to assess the performance of the Influx sorter (Becton Dickinson) to separate single viruses from a viral culture before working with natural viral samples. To prepare bacteriophage cultures, P1 was grown as previously described[41] and then, the culture was centrifuged at 6,000g for 15 min, and the supernatant filtered through 0.22 μm syringe polyethersulfone (PES) membrane filters (ref. SLGP033RS, Millipore, Milford, MA, USA) to purify the viral fraction. The presence of bacteriophage P1 was confirmed by nucleic-acid staining and epifluorescence microscopy[42] before flow cytometry analyses and sorting.

**Virus staining optimization for flow cytometry analyses.** Standard protocols for detecting viruses by flow cytometry are performed typically on fixed samples with 0.5% of glutaraldehyde[32], which ultimately for our purpose would prevent the amplification of genetic material by multiple displacement amplification (MDA). In this work, we carried out SVGs with unfixed and fixed samples. For fixed viral samples, they were first 0.2 μm-filtered and then fixed with 0.1% of glutaraldehyde final concentration and processed as described in detail[20] with the exception that the used dye was SYBR Gold to 0.5 × final concentration (Invitrogen catalogue no. S11494). For staining unfixed viral samples, the protocol was as follows. SYBR Gold commercial stock with a concentration of 10,000 × was diluted to 1,000 × in sterile MilliQ water, filtered through 0.02 μm Anotop filters (Whatman, ref. 6809–1002) and stored at − 20 °C in the dark. Viral samples (typically 1 ml), previously filtered through 0.22 μm syringe PES membrane filters, were concentrated to 50 μl with Nanosep 10 kDa (OMEGA, Pall Life Sciences) and washed with 500 μl of sterile 0.02 μm-filtered TE buffer (10 mM Tris, 1 mM EDTA; pH 8.0) to remove free DNA. The viruses in sterile TE buffer were then stained with SYBR Gold (final concentration of 4 ×) at room temperature for 20 min in the dark and washed three times with 500 μl of sterile 0.02 μm-filtered TE buffer in the ultracentrifugal devices. Finally, 500 μl of sterile 0.02 μm-filtered TE buffer were added to the column and recovered for flow cytometry analyses and sorting. The whole staining procedure was applied to blanks for flow cytometry analyses as per recommendation of reference viral staining protocols[32] to identify the correct viral gates for analyses and sorting. A similar staining protocol for fresh unfixed samples has been successfully used for flow cytometry to stain *Synechococcus* phages[34]. We noted that SYBR Gold provided better resolution than SYBR Green for unfixed samples.

Optimization of virus staining previous to flow cytometry sorting was performed with a FACS Canto II cytometer (BD Biosciences) equipped with a 488-nm laser. A threshold was set on green fluorescence at a value of 200, and samples were analysed using a flow rate below 1,000 events per second to avoid coincidence of viral particles[32]. Green fluorescence, total counts and side scatter were recorded for 1 min for each analysed sample and blank.

**Fluorescence-activated virus sorting.** BD Influx sorter (Becton Dickinson, San Jose, CA), reagents and disposable material for sterile FAVS were DNA decontaminated as described in detail[43] with some modifications. Sterile TE buffer used for staining viruses was previously ultraviolet-treated for 16 h in a UVP Ultraviolet CL-1000 Crosslinker. 384-well plates (ref. 4ti-0384; 4titude Limited, UK) were autoclaved and then 0.6 μl of ultraviolet-treated 1 × TE buffer was added per well. Plates were then ultraviolet-treated for 10 min without a cover (≈ 10 cm distance from ultraviolet lamps) in a laminar PCR hood (Alpina K1000) equipped with three ultraviolet lamps (18 W × 3) that sterilize the incoming flow air. Finally, once the plate was set in the Computerized Cell Deposition Unit (CCDU) of Influx sorter was again ultraviolet-irradiated for at least 2 min. Before virus sorting, stained samples were pre-screened through a 35-μm mesh-size cell strainer (BD Biosciences). BD Influx jet-in-air cell sorter was selected for FAVS because of the fine-tuning and high-resolution capabilities. The instrument was equipped with a high-power blue 488-nm laser at 200 mW that was set to 100% power to improve nano-particle detection such as viruses with very low fluorescence emission signal.

In addition, the Influx sorter is equipped with a small-particle detector in which a forward scatter detector is replaced for a high-performance photomultiplier tube (PMT). Furthermore, a mechanical diaphragm, working as a pinhole, was used for fine laser alignment to maximize fluorochrome excitation and fluorescence collection. Before virus sorting, instrument setup was performed using standard 8-peaks Rainbow beads (Sphero Rainbow Calibration Particles 3.0–3.4 μm, BD Biosciences, ref. 559123) for laser alignment. In addition, 220 nm 1-peak yellow beads (Sphero Nano Fluorescent Particles, Yellow 0.22 μm, Spherotech Inc., ref. NFPPS-0252-5) were used for instrument fine-tuning to obtain highest resolution of nano-particle detection. For virus sorting, instrument was set to 'Single' sort mode, which is the most rigorous setting to sort single particles. For sorting, threshold on green fluorescence was set at 1.0 for detecting SYBR Gold fluorescence through a light line passing a 505 LP filter and collected by 530/40 nm band-pass filter. The 100 μm nozzle was chosen because of the best piezoelectric-frequency/electronic-noise ratio with the piezoelectric frequency adjusted at 38.7 kHz. In addition, 100 μm nozzle can work at relative low pressure (20 p.s.i.) compared with 70 μm nozzle (40 p.s.i), reducing particle speed with a consequent increase of exposition time of a particle passing through a laser beam (time-of-flight), which ultimately allowed to collect more fluorescence signal per stained viral particle. Initially, electronic noise without sample acquisition was detected to set the baseline for fluorescence signal detection. For that, green fluorescence PMT voltage and trigger was adjusted to 20–30 events per second with a low sample differential of 1.0 p.s.i. approximately. Then, blanks as described above were analysed to aid in gate selection for virus sorting with a similar sample rate (20–30 events per second) as that of electronic noise. Since very low sample flow rate is mandatory for single-virus sorting to prevent doublets, virus sample flow rate was adjusted to 40–50 events per second. Considering that 20–30 events could come from electronic noise and that a total of 38,700 drops per second were generated (piezoelectric at 38.7 kHz), the estimated ratio of putative-sorted particle versus generated drops was 1/1,300, which ensured a separation enough between each sorted particle to prevent sorting of doublets. All parameters (forward scatter, side scatter and green fluorescence) were collected in logarithmic mode and analysed with BD FACS Software, version 1.0.0.0.650 (Becton Dickinson, San Jose, CA). Fine alignment of 384-well plates in the CCDU was performed by visually inspecting the colour change of small disks (1 mm size) of litmus paper placed at the bottom of the wells, caused by the deposition of sorted droplets. Layout of 384-well plates for viral samples was as follows: 332 were dedicated for single viruses, 44 were used as negative controls (no droplet deposition), 2 received 10 viruses each, 2 received 20 viruses each, and 4 wells were used as positive controls with 1 ng of genomic lambda DNA (New England Biolab). Plates were then covered with sterile film and stored at −80 °C until used.

**Confocal microscopy of single viruses.** Imaging of sorted P1 bacteriophages was performed on a TCS SP5 II CW-STED Leica microscope equipped with a Leica Confocal Software (LasAF 2.5.1) at the Centre for Genomic Regulation (CRG) (Barcelona). For imaging of single viruses, individual viral particles were sorted directly on a slide and scanned using a HC PL APO × 100/1.40 oil objective with a 488 nm argon laser line (power to 33%), high-disk adjusted to 200% and the PMT 2 gain to 700 V.

**Marine sample collection and processing.** SVGs was performed for the following collected samples: (i) surface seawater from the Blanes Bay Microbial Observatory (BBMO) in the north-western Mediterranean Sea (41°40′13.5" N 2°48′00.6" E; 2.7 miles offshore) collected on 15 April 2015 (chlorophyll a concentration 0.32 μg l$^{-1}$ and temperature 14.6 °C), (ii) surface seawater samples from the Barcelona Beach (Barcelona, Spain, 41°23′01.7" N 2°11′50.0" E) collected on 19 November 2014 (iii) mesopelagic seawater sample taken in the South-Western Mediterranean Sea (37°21′12.96" N 0°1710.32" W) from the deep chlorophyll maximum (DCM), 60 m depth, on 15 October 2015 and (iv) deep water samples collected in the North Atlantic Sea with the Malaspina expedition, at stations 131 (17°25′39"N 59°49′43" W) and 134 (18°19′38" N 52°38′20.15" W), on 26 November 2011 and 29 November 2011, respectively, both from 4,000 m depth. Metadata for station 131 is: $T\Delta$. 2.31 °C, prokaryote abundance of 2.8E$^4$ cells per ml and virus-like particle abundance of 1.3E$^5$ VLP per ml. Metadata for station 134 is: $T\Delta$ = 2.26 °C, prokaryote abundance of 2.41E$^4$ cells per ml and virus-like particle abundance of 1.14E$^5$ VLP per ml.

Surface and DCM seawater samples were immediately filtered through 0.22 μm syringe PES membrane filters (ref. SLGP033RS, Millipore, Milford, MA, USA). Surface samples were processed for FAVS (see above) within the same day, while the DCM sample was conserved at 4 °C until the sorting on 5 November 2015. The Malaspina expedition samples were cryopreserved as described[12] until sorting.

BBMO metavirome and microbial metagenome were constructed in this study as follows. For microbial metagenomics, 100 ml of seawater was filtered through 0.2 μm filter (ref. SLGP033RS, Millipore, Milford, MA, USA) and nucleic acids extracted with MasterPure Complete DNA and RNA Purification Kit '(Epibio, Illumina) according to the manufacturer's protocol.

For seawater viromics, 28 l of seawater was sieved through a 20-μm mesh, filtered through a 0.2 μm filter (ref. SLGP033RS, Millipore, Milford, MA, USA), and then viruses were concentrated to 20 ml from the filtrate using tangential flow filtration with a 30 kDa polyethersulfone Vivaflow 200 membrane (Sartorius).

The virus concentrate was again 0.2 μm-filtered to ensure that no cells remained, which was later confirmed by SYBR Gold staining by epifluorescence microscopy as described[42]. Then, the viral fraction was concentrated to 1.5 ml with Amicon Ultra-15 (Millipore), washed with 10 ml of sterile TE buffer to remove free small DNA fragments, and then 1.5 ml of viral concentrate was treated with 2.5U of Turbo DNase I (Ambion) at 37 °C for 1 h to remove the remaining free DNA. Finally, the viral fraction was ultra-concentrated to 150 μl with Amicon Ultra-50 (10 kDa-cut off, Millipore) and nucleic acids extracted with MasterPure Complete DNA and RNA Purification Kit (Epibio, Illumina) according to the manufacturer's protocol. PCR amplification for 16S rRNA gene with primers 341F and 907R (ref. 44) with the cycling conditions as described[45] was not obtained from the extracted viral DNA, which indicated that contamination with bacterial DNA is negligent.

**Whole-genome amplification of single viruses.** MDA procedure and DNA decontamination of reagents prior to MDA set up was done as described[43] with some modifications. Single-virus 'lysis' was done by a combination of liquid N$_2$ shock and/or cold KOH lysis. First, upon thawing 384-well plates at 4 °C, plates were carefully immersed in liquid N$_2$ for 30–60 s. avoiding contact of liquid N$_2$ with the film covering the plate. Then a quick thawing shock was applied in a 45 °C water bath for ≈1–2 min. This cycle was repeated two to four times. Next, to each well, 0.7 μl of lysis buffer D2 (see details for preparation and composition in ref. 43) was added and incubated for 5 min at 4 °C. KOH lysis reaction was stopped either with 0.7 μl of Tris-HCl pH 4 or 0.7 μl of Stop solution (Qiagen, ref. 1032393) per well. Then, genomic DNA from the lysed single viruses was amplified by MDA in a 10 μl final volume reaction. The master mix MDA reaction contained 0.26 μl of phi29 DNA polymerase (ref. M0269L; 10 U μl$^{-1}$; New England Biolab), 1 μl of Phi29 10 × reaction buffer (ref. M0269L; New England Biolab), 1 μl of hexamers (0.5 mM; IDT), 0.1 μl of DTT (1 M; Sigma), 0.4 μl of dNTPs (10 mM each; ref. N0447L, New England Biolab), 0.002 μl of SYTO 9 (Invitrogen) and 5.2 μl of sterile ultraviolet 16 h-treated mQ water. The MDA master mix, except SYTO 9, was ultraviolet decontaminated for 15 min at 4 °C in a UVP Ultraviolet CL-1000 Crosslinker as described in detail[43]. After ultraviolet treatment, SYTO 9 was added to the master mix. Finally, 0.6 ng of genomic lambda (ref. N3011S, New England Biolab) was added to wells A1, A24, P1 and P24 of the plate as positive control. MDA reactions were incubated at 30 °C for 16 h in a CLARIOstar plate reader (BMG Labtech) to monitor the whole-genome amplification. The MDA reaction was stopped by heat-inactivation of the phi 29 at 65 °C for 10 min and the MDA product was diluted 50-fold in sterile TE buffer. Overall 0.5 μl aliquots of the dilute MDA products were served as templates for PCR screening of 16S rRNA gene to assess exogenous bacterial contamination. PCR amplification was performed with primers Prok_340F and Prok_806R as described[46]. No amplification was obtained for the single amplified viral genomes. The MDA Cp values indicated time (hours) required to reach half of the maximal fluorescence in each well. Mean Cp values for positive controls with 0.6 ng of total DNA per well was ≈4–6 h, while Cp MDA values for positive single amplified viral genomes was ≈10–11 h.

Subtle variation of phi29 polymerase activity (ref. M0269L; 10 U μl$^{-1}$; New England Biolab) has been detected during this study across different batch numbers of enzyme that affect Cp MDA values. As the amount of DNA template from single viruses is significantly less than for single bacterial cells, we recommend to test the activity of phi29 ahead with 0.6 ng of lambda DNA (positive control) template to obtain the above mentioned Cp values to guarantee enough a priori activity to amplify genetic material from single viruses. At the same time, ultraviolet decontamination step has to ensure little or no background amplification in the negative controls. The concentration of enzyme used in this study has been 0.26 μl per well, but notice that, the amount of enzyme and ultraviolet decontamination should be adjusted accordingly to obtain the above-mentioned Cp values for positive controls as described[43].

**Evaluation of free DNA in sorted seawater microdroplets.** Seawater sample from BBMO was sequentially filtered through 0.2 and 0.02 μm filter. The elute, free of viruses, was then stained as above, with the exception that no fixation and liquid nitrogen shock was applied to avoid degradation of putative-free DNA. Flow cytometry sorting of single sorted events from the putative-stained fraction and MDA reactions were performed to evaluate the putative presence of free DNA in the seawater volume co-occurring with single sorted viruses in the microdroplet.

**Sequencing and genome analyses of single-viruses.** Single-amplified viral genomes, microbial metagenomes and viromes were sequenced by Illumina technology using the Nextera XT DNA library (ref. FC-131-1024, Illumina) in a MiSeq sequencer (2 × 250, pair-end) according to the manufacturer's protocol. In addition, four vSAGs (37-I21, 17-E11, 37-F6 and 37-L15) were also sequenced by using TruSeq DNA PCR-Free library (ref. FC-121-3001) in a MiSeq sequencer (2 × 150, pair-end) according to the manufacturer's protocol. The reads were quality-filtered using prinseq-lite program[47] with the following parameters: min_length: 50, trim_qual_right: 20, trim_qual_type: mean and trim_qual_window: 20. Genome assembly was performed with SPAdes version 3.6.1 (ref. 48) by applying the following parameters: --sc, -k 33,55,77,99,127, --

careful. Generated contigs were subjected to another round of assembly using Geneious R8 bioinformatic program[49] with stringent conditions (100% sequence identity in the alignment, no gap and a minimum of 200 bp of overlapping). Then, a thorough manual inspection was done for all resulting contigs to ensure a non-chimeric assembly. Specifically, we reviewed those merged contigs resulted from Geneious post-assembly, one at a time, to corroborate that no mismatches were presented in these contigs. Contigs <1,000 bp and contigs matching to human DNA or common bacterial contaminants in SCGs and sequencing[50,51] were removed from the analyses. Contamination screening was done by using a combination of ProDeGe program and a comparison with the database nr/nt using the stand alone BLAST version 2.2.31 + . Finally, prediction of open reading frames (ORFs) from the curated viral genomes and genome annotation were done in Metavir platform by using the default parameters as described[21]. In addition, in parallel predicted ORFs from Metavir were also compared in house by BLASTp with version 2.2.31 + against the non redundant (nr) database (date 28 October 2015) with the following parameters: e-value <1e − 5. We obtained similar annotation than that from the Metavir platform.

For vSAG 37-F6, five specific primer (Supplementary Table 8) sets covering different genomic regions were designed and successfully tested for the corresponding MDA product of that single viruses, which validate the results from genome assembly. Prediction of structural proteins was conducted with an artificial neural network algorithm[52]. The Metavirome from BMMO was assembled by IDBA-UD using option '-precreation'[53], while the BBMO metagenome with SPAdes 3.8.1 using metaspades options with parameters -k 33,55,77,99,127. Contigs were annotated as above. Microbial taxonomy profiling for the metagenome from BBMO was carried out with riboFrame[54]. Whole-genome alignment was first performed with Mauve program[55] and polished with CLUSTAL W aligner and finally manually inspected. Alignment identity values for each nucleotide position was calculated in Geneious bioinformatics software[49]. Calculation of average genome nucleotide identity (ANI) among viral genomes was calculated with the Gegenees software with the following parameters: fragment size = 100 and step size = 50 (refs 56,57). Alignment of large subunit of terminase (TerL) and phylogeny was carried out as described[10,11]. Protein alignment was done with CLUSTAL W implemented in Geneious bioinformatics package. Prediction of 3D structure of capsid proteins was carried out as described[31] with the online server i.-Tasser. In silico digestion of predicted proteins from vSAGs was performed with the on-line bioinformatics tool PeptideMass in ExPASy resource portal (http://web.expasy.org/peptide_mass/). Parameters used for searching the predicted proteins of vSAGs was as previously described[31]: parent mass tolerance, 3.0; fragment ion tolerance, 0.5; up to four missed cleavages allowed, variable modification of carboxymethyl cysteine ( + 57.021 Da) and tryptic peptides only.

**Gene-content-based network analysis.** Proteins were predicted from the marine vSAGs (61 sequences, 1,192 proteins, respectively) using metagene annotator[58], and added to all proteins from bacterial and archaeal viruses from NCBI RefSeq (2,010 sequences, 198,102 proteins, v75), from predicted proteins from the GOV data set (370,165 proteins[59]), and from environmental phage from Genbank (40,803 proteins). This resulted in 610,262 proteins from 17,744 sequences. Proteins were compared through all-verses-all BLASTP with an E-value threshold of 10 − 5 and 50 for bit score. Protein clusters (PCs) were then defined using Markov Clustering Algorithm (MCL)[60], using default parameters and 2 for an inflation value. vContact (https://bitbucket.org/MAVERICLab/vcontact)[61,62] was then used to calculate a similar score between every pair of genomes based on the number of PCs shared between two sequences and all pairs using the hypergeometric similarity, as previous[2,59]. MCL was applied to the similarity scores using a threshold of 1 and MCL inflation of 2 to generate viral clusters (VCs, ≥2 sequences). A total of 933 VCs (17,149 sequences) were obtained, with 31 containing at least one vSAG. Sequences were post-vContact analysed using custom python scripts that performed the following functions: identification of highly similar VCs from the GOV data set using the Jaccard similarity (with the highest similarity values used to associate VC members of vSAG-VCs and GOV-VCs), predicted taxonomy using reference sequences present within the VCs (below), and constructed a network (the python package network[63]) using the similarity scores generated by vContact between each genome pair. Taxonomy predictions were based on the presence of reference sequences within each VC, with either (1) a 'majority-rules approach' where the most abundant (≥ 50%) reference sequence taxonomy being applied to all VC members (that is, if 60% of reference sequences were Caudovirales, then the entire VC was classified as such) or (2) using a 'lowest common ancestor' approach among the reference sequences, where taxonomic lineages for each reference within the VC were compared to identify the lowest taxonomic rank (order, family, genus and so on) that contains all the reference sequences. To reduce the complexity of visualizing 17,149 sequences, network components (node groups disconnected from other node groups) not including at least one vSAG were excluded. The final data set (sequences) was exported to Cytoscape (v3.3.0)[63] and images were post-processed using Adobe Illustrator CC 2015.

**Metagenomics and metaproteome fragment recruitment.** To estimate the abundance and distribution of marine vSAGs, we performed a comprehensive fragment recruitment using different marine metagenome and virome data sets

from the Tara Oceans expedition[3], Pacific Ocean Virome[23], the Sargasso Sea[24] besides those generated in this study from the Mediterranean Sea. Viromes from the deep ocean were from the Malaspina Expedition and are publicly available at the Joint Genome Institute (see ref. 31 for details). In addition, to compare the in silico abundance of our vSAG, the following reference marine viruses were included in the analyses: 5,468 viral contigs from surface and DCM (≥10 kb) from the Tara expedition[3], viral genomes obtained from single cells (the longest contig for each virus)[26], 179 marine virus isolates available at IMG database (Supplementary Table 9) and marine viral genomes reconstructed from fosmids[10,11]. Fragment recruitment analyses were carried out with stand alone BLAST version 2.2.31 + similarly as described[10] but with an e-value <1e − 5, and a query coverage >80%. The used commands were as follow: 'blastn -db Viral_data_base.fasta -outfmt "6 qseqid sseqid salltitles sallseqid pident bitscore evalue length qstart qend sstart send" -out recruit_name.txt -query Virome.fasta -evalue 0.00001 -perc_identity 50 -num_threads 6'. Then, by using R software, two different identity percentage cut-offs were applied, 'perc_identity 95' and 'perc_identity 70' (Supplementary Note 5), to recruit only reads from putative closely related viruses or reads from distantly related viruses as well. R software was also used to remove hits with query coverage <80% and normalize according to genome and metagenome size,as in the previous studies[3,10] to estimate the kb recruited per kb of genome per Gb of metagenome (KPKG). One-way ANOVA was calculated in R package by using the viral data set as factor. Alternatively, reciprocal best-hit fragment recruitment with the viral data set was done as described[27] but employing the Enveomics bioinformatics package (https://peerj.com/preprints/1900/). For the metaproteomic recruitment analyses, peptides obtained from the Oregon coast[64] and Tara Oceans Expedition[31] were compared with the vSAG and the above-mentioned viral genomes from different data sets by using BLASTx and BLASTp with the optimized parameters for short sequences as manual describes. Recruitment data were normalized according to viral genome size. Those peptide signatures matching 100% sequence identity and coverage with translated ORFs of vSAG were also screened to assess geographical distribution along the different metavirome data sets.

**SNPs of vSAGs in viromes.** To estimate the frequency of SNPs of vSAGs at the level of viral species, first we mapped the virome reads from the Mediterranean Sea MS022 and the BBMO, which was the sampling site where vSAGs were obtained. As sequencing errors could bias the SNPs analyses, it is important to remark that only those raw reads from BBMO and Tara viromes passing the quality filtering were considered. For Tara viromes, parameters for quality filtering[3] were similar to those used here for BBMO virome (see above) since as previously described[3], reads were removed when the median quality score was <20 and bases were trimmed at the 3′ end of reads if the quality score was <20. For the Tara MS022 virome, the two most abundant TOV and GOV viral contigs were used. For the BBMO virome, the 25 best viral contig recruiters that ranked within the top 30 in the recruitment were considered. For vSAGs, a total of five best recruited, within the top 30 best recruiters, were considered. For SNP calculation, first virome reads were mapped by using Geneious bioinformatic program[49] against the reference viral contigs and vSAGs with the following parameters: ≥95% nucleotide identity, ≥70% of read coverage and sensitivity 'fast/read mapping'. SNP calculation with the mapped reads was carried out with Geneious bioinformatics program with the default parameters except for the coverage, that a minimum coverage of 5 × was considered. In fact, for most of the obtained SNPs the observed coverage was >20 ×. In all cases, the detected SNPs had a P value <0.000001 (binominal coefficient implemented in Geneious bioinformatic package under tool 'Find Variations/SNPs'), which also considered the probability of a nucleotide variant because of sequencing errors. The 'effective' genome fraction (kb) with a minimum of 5 × coverage was considered for the normalization of SNPs frequency. Thus, the estimation of frequency of SNPs for vSAGs and viral contigs was number of SNPs per kb of effective mapped viral genome per 1 Mb of mapped virome reads.

**Simulation of viromes with different microdiversity degrees.** The genome assembly performance of the assemblers IDBA_UD[53] and SPAdes[48] for natural virome data sets with populations with different degrees of diversity and microdiversity was assessed. While recently, IDBA_UD has been commonly used for metagenomic assembly.

SPAdes, with the new version optimized for metagenomics (option 'metaspades') is currently considered as one of the most powerful assemblers to address uneven sequencing genome coverages typically obtained in a metagenome (see link: arXiv:1604.03071). The general method for simulation of natural viromes with different degrees of microdiversity for vSAG 37-F6 is depicted in Supplementary Fig. 20 and explained in detail in Supplementary Methods.

**Data availability.** Raw sequences of metagenome and metavirome from BBMO sample have been deposited in the European Nucleotide Archive under the accession number PRJEB12379. Genomes of vSAGs have been deposited in Genbank under accession numbers KY052794–KY052854. Genome annotations are in JGI-IMG under GOLD ID projects Gp0155348–Gp0155387, and Gp0155393–Gp0155396. Genomes of vSAGs and annotations, as well as metagenome viral contigs assembled from the BBMO virome have been

deposited in Cyverse and are publicly accessible with the following link: http://de.iplantcollaborative.org/dl/d/288CCEA1-7C16-47FA-9E60-5628B695D842/vSAGs_Data.zip. Data of simulated genomes and *Tara* MS22 virome with vSAG 37-F6 populations with different degrees of microdiversity are publicly available with the following links: http://de.iplantcollaborative.org/dl/d/0D119B9B-D912-4D1A-8554-F27FCF3F6E8A/SIMULATION_VIROME_TAR-A22.zip and http://de.iplantcollaborative.org/dl/d/5D6E8373-6864-4303-8073-8A7A25B4ADDB/Simulated_genomes.zip. Data on relative recruitment frequencies for each viral data set and relatedness of vSAGs with nearest viruses within VCs from GOV data set at the protein level is available in Cyverse with the following link: http://de.iplantcollaborative.org/dl/d/0BA309BE-980E-42AC-A935-2CB564E1F91C/Virome_Recruitment.xlsx. All other data are available from the authors upon request.

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

## Acknowledgements

This work has been supported by Spanish Ministry of Economy and Competitiveness (refs CGL2013-40564-R and SAF2013-49267-EXP), Generalitat Valenciana (ref. ACOM/2015/133 and ACIF/2015/332), the USA National Science Foundation (OCE#1536989), the USA Department of Energy (DE-SC0010580), and Gordon and Betty Moore Foundation (grants 3305, 3790, and 5334). The Ohio Supercomputer supported gene-sharing network high performance compute time. Work at BBMO was funded by Spanish project CT2015-70340-R. Work at CRG, BIST and UPF was in part funded by the Spanish Ministry of Economy and Competitiveness, 'Centro de Excelencia Severo Ochoa 2013-2017' and the Spanish Ministry of Economy and Competitiveness, 'Centro de Excelencia Maria de Maeztu 2016-2019'.

## Author contributions

M.M.-G. conceived and led the study. F.M.-H. led the analyses and interpretation of data. M.M.-G., F.M.-H., O.F., M.L.G., B.B., M.J.d.l.C.P., J.M.M., J.A., J.M.G., R.R., F.R.-V., M.B.S. and S.G.A. participated in the analyses and interpretation of data. M.M.-G. and F.M.-H. wrote the paper.
