## [Peer Review File · Nature Communications]

Reviewers' comments:

Reviewer #1 (Remarks to the Author):

Overall this is an exciting piece of work that describes the use of single amplified viral genomes for the purpose of investigating viral diversity and its role in microbial ecology of the oceans and human habitats. This work is quite interesting and the technology exciting and at the cutting edge. However, the article is also very technical. The methods are very long and yet highly important for this work. In fact, much of the novelty of this work are the methods themselves, and yet they are not thoroughly discussed in the main text. I think the authors did themselves a disservice by not including more of the advances in the method in the main text.

Further the authors seem to rush through the work in the text and skip over many important and interesting aspects of the work. Overall the work feels hurried and incomplete in its current form. Thus I recommend a major revision where the authors spend significant time rewriting the work for a broader audience and include introduction and discussion sections that more informative.

It is my suspicion that based on the seemingly rushed writing and awkward format that this was originally a Nature rejection that received a slapped together and boiler plate intro and discussion to conform to the Nat Com format. To be acceptable and relay the truly important and novel aspects of this work, the main text should contain: a concrete hypothesis, an explanation of the advancement in the single amplified virus genome approach (this is the best part), details of the reasoning for the experimental design, and a clear and complete explanation and discussion of the data.

In its current form this work is too narrow and misses an opportunity to really inform the community at large about the utility and limitations of this approach or what these viruses are or doing (other than being abundant).

Lastly, it is not clear how the increase in read lengths of these viral genomes contributes to our understanding of viral microdiversity. By my reading, it seems that what you found was in fact similar to other past works. Those networks reveal that your viruses are within the groups previously found but that some aspect (which I guess I missed) makes them more diverse at the nucleotide level. This seems to be the main novel finding yet it's not entirely fleshed out.

Reviewer #2 (Remarks to the Author):

In, "Single-virus genomics reveals hidden cosmopolitan and abundant viruses", the authors developed a method to sequence single viral particles. They sorted and genomically amplified a total of 2,658 viral particles, for which 47 were randomly selected for genome sequencing. The aim of this study was to demonstrate that single cell viromics (SVG) is an

unbiased way, as compared to metagenomics and culturing, to look at viral communities.

The manuscript is overall well written, but there are some sentences that need revisions and editing. A few examples:

Line 86: ..."in spite of a handful..."

Line 123: ..."putative hosts did not clarified..."

In this study, they looked at single viral genomes from seawater and saliva samples. The saliva sample, as currently written in the manuscript, seems like an after thought and do not bring anything to the story, except confusion. There is only a short paragraph at the end discussing the single interesting virus that was found in saliva, with no figure related to it. It should either be removed completely, or more details should be added to discuss in depth the novelty of the presented method in the field of human viromics.

The microdiversity simulations are compelling and aim to answer a very important question in the field of environmental virology: why are assembled genomes in viromics not the most abundant viruses? This is a very important point that the authors make and should be emphasized. One thing that the authors did in their analysis, but did not present in the manuscript, is how biogeography impacts viral distribution and microdiversity. It would be interesting to choose a few of the most abundant viruses and show how their abundance changes depending on the virome used for metagenomic recruitment.

Single virus genomics has potential to look at viral diversity, as demonstrated in the manuscript, but also has disadvantages, which should be discussed, or at least pointed out in the manuscript. For example, no link to the host can be done, which limits ecological impact interpretations. On lines 122-123, the authors mention that they attempted to identify the host of the virus 37-F6 using tetramer frequency (Suppl. Fig. 8). This analysis is very weak and unnecessary. First, tetranucleotide frequency is not related to the host, and second, the other sequences consist only of a few related viruses. Obviously, closely related viruses will have more similar tetranucleotide frequencies. There are a few ways to link environmental viruses to their environmental hosts, which include looking at CRISPR inserts or tRNA genes.

On line 92, the authors again discuss the potential ecological impact of viruses by suggesting that they were actively infecting marine bacterioplankton. Single viral genomics in no way can determine if viruses are active. This sentence is pure speculation and should be either more detailed or removed.

Another disadvantage of SVG is that it is very expensive, therefore we can only look at a few viral genomes at a time, rather than the whole community as in viromics. The authors briefly mention that their dataset is limited on line 86, but didn't go further.

On line 64, the authors mention that the sample size necessary for SVG is very small (1 ml). Because of the small sample size, and the low number of genomes sequenced (47), the authors should be more cautious when discussing that they sequenced all the major viruses and viral groups.

Reviewer #3 (Remarks to the Author):

Summary: The authors of this manuscript build on previous work where they had been able to successfully apply single cell genomics technology to virology. Here they take one step further and demonstrate these capabilities in environmental samples, a remarkable feat on its own. They were able to sequence 47 viral genomes (from marine and salivary samples) with different degrees of completeness; after a series of bioinformatics analysis they assigned taxonomy and clustered with previously known clusters of viral genomes. To place their findings in an ecologically relevant framework the authors then assessed the abundance of their viral genomes in different metagenomic and proteomic datasets, and the authors argue that their newly discovered viruses are among the most abundant in these global datasets. Interestingly, the authors note that one of these abundant virus 37-F6 had only loose similarity to a few viral-like contigs recovered from uncultured sources. These two contrasting observations are seemingly at odds: abundant in metagenomic samples yet barely ever seen, so the authors argue that it might be the high levels of intra-population variation that hinders assemblability of such abundant populations (I agree it is a good possibility, but a simulation would be more convincing). Finally the authors extend their proof of concept to another environment, human saliva. While there is a great deal of valuable information here and I applaud the approach in general as a significant advance in the field, I have several concerns about details of the analysis that are critical for the conclusions.

General criticisms:

The use of single cell technology had been previously adapted to virology, in the present case it is a remarkable improvement to be able to use in environmental samples. However, I am afraid that in the process of working to make their findings ecologically relevant, the authors apparently overlooked some important details in their bioinformatics analysis (specific details further down below):

1) Fragment recruitment; Nowhere in the methods did I see details of what kind of recruitment was used; to avoid difficulties I hope it was at least "competitive" (if not reciprocal best hit) and that it was done at the nucleotide level. The details are incredibly important to interpretation, and this lack of critical information makes evaluation difficult. Maybe it is fine. I suggest the authors should use the methods described in references 1,5 and 9.

2) Peptide recruitment; This one is very troubling, as proteomic peptides should not be aligned against a reference just with blast; Two fundamental things are different from nucleotide sequences: 1) These peptides are not random, as there is an enzymatic digestion that cleaves proteins at specific (and known) sites. 2) These are very short, often <10 AA residues long. These peptides are used to identify longer proteins based on multiple hits and non redundant matches; the latter is particularly important as peptides are short and will often be covering a domain that can be represented in multiple sequences. Additionally, when identity is lower, it can be matching only a few amino acid residues (potentially misleading, and databases are sparse compared to global diversity).

As the authors appear very keen on using proteomic data, I would recommend they use the proteins themselves that were identified with these peptides in the original published work and not the peptides alone. Reference 28 identified proteins as structural, and such annotation was later propagated by the use of protein clusters within the Tara dataset. Also, some work used specialized spectral searchers such as Xtandem, Percolator or Sequest.

Specific comments:

LINE 13: I believe the authors mean either 44, or 37, since it only becomes 47 after adding the salivary viruses.

LINE 57: Most marine vSAGs... This sentence would benefit from including and N-number.

LINE 97-101: The findings described here are exciting but I think that it might not be fully supported by the data; Please see below for further explanation on the metaproteomics analyses.

LINE 102-112: Readers will benefit from clarification on the kind of recruitment done by the authors (see general criticisms). Reference 1 includes a great description of the methods best suited to recruit metagenomic reads against a few novel references. The authors of 1 have a sensitivity analysis showing a decrease of recruitment as more closely related sequences are added. Notably the methods of 1 have been used in 5 and 9. It seems to me that the best way to analyze this would be reciprocal best blast hit as in references 1,5 and 9. Perhaps that was done here, but it needs to be clarified.

LINE 146-154: It is unclear to me what the saliva viruses are adding to the story. Needs a justification beyond the fact it was done.

LINE 182-206: The proteomics figure will need revision.

Methods Section:

LINE 468: How did the authors look for chimeric assembly (a significant potential problem)? Can you describe the methods and parameters used?

LINE 473: I assume the authors used a specific set of parameters with metavir? If so, please state them.

LINE 473-474: How did the authors predict Open Reading Frames? The methods skip directly to annotation without describing how genes were predicted. Please state the program and parameters used.

LINE 474-474: The authors should describe exact settings used during their blast comparison to RefSeq Virus. Additionally, what kind of blast was used? Why did the authors use only the viral portion of RefSeq? Does that "stack the deck" or add bias?

LINE 507: Can the authors please formalize "most prevalent" to an exact notation?

LINE 527: Can the authors describe why they only used the longest contigs during their recruitment work? Justification for excluding the shorter ones, and a particular cutoff?

LINE 530: As noted above, it is not clear if the recruitment work was done competitively or not, i.e. is each read only mapped once? Since this is a central finding, arguments about abundance and prevalence are based on results from it, the authors are recommended to use the methods in Reference 1, 5 and 9. Whatever they used, the authors should describe it more clearly. If not competitive or reciprocal best hit, it could be problematic.

LINE 534: Reference 60 is not appropriate, in that work uses TEM data and not any kind of -omics; Perhaps the authors meant reference 5, with the same lead author?

LINE 539: I believe Tara Oceans Expedition, not Malaspina.

LINE 538-543: Proteomics analyses should not be done with the use of the peptide recovered from the referenced work. There are fundamental differences from metagenomic reads: 1) Peptides are short 2) They are not random.

Regarding the second point, it is not clear if the authors ensured that the area matching these peptides within their genomes was forced to be flanked in the C-terminal by a Lysine or Arginine (for example from data from reference 28 where trypsin was used in the digestion).

These peptides are recovered by matching different spectra against a reference database; when multiple peptides can be identified we can only be sure a protein was identified if a protein has both multiple spectra associated to it and if it has unique non redundant spectra associated to it. The latter point is especially important as peptides could often, due to their short nature, cover only a part of a domain that repeats in multiple proteins. Not a small issue.

If the authors wish to use proteomic data, two alternatives are recommended:

- 1) To use the proteins that the authors of 28 and 29 identified from these spectra (or peptides) and blastp them as one normally would in the case of protein sequences.
- 2) To search all the spectra with specialized searchers (xtandem, sequest, percolator) including their own genomic sequences in the database + plus whatever it was used by the works 28 and 29.

LINE 544 and after:

Methods do not include details on the construction of the phylogenies and they are not referenced within the main text. The authors should note details on the construction, program, algorithm, etc.

LINE 552: The authors should provide details on the kind of QC that the reads had before going into alignment for SNP calling. QC is described for reads coming from their own work (a few sections before) but not for reads taken from Tara, for example. Such details can

matter and should be included.

We would like to thank the reviewers for their work and effort on this manuscript. We are convinced that the new version of the manuscript has been improved after addressing all their concerns. All changes and edits are highlighted in color in the new version of the manuscript. The previous version has been substantially modified and updated according to their concerns and suggestions.

Reviewer #1 (Remarks to the Author):

Overall this is an exciting piece of work that describes the use of single amplified viral genomes for the purpose of investigating viral diversity and its role in microbial ecology of the oceans and human habitats. This work is quite interesting and the technology exciting and at the cutting edge. However, the article is also very technical. The methods are very long and yet highly important for this work. In fact, much of the novelty of this work are the methods themselves, and yet they are not thoroughly discussed in the main text.

I think the authors did themselves a disservice by not including more of the advances in the method in the main text.

We appreciate the time of this referee on our manuscript. The advances in the methods are now addressed in the main text, mainly in the discussion section.

Further the authors seem to rush through the work in the text and skip over many important and interesting aspects of the work. Overall the work feels hurried and incomplete in its current form. Thus I recommend a major revision where the authors spend significant time rewriting the work for a broader audience and include introduction and discussion sections that more informative.

We have re-written our manuscript for a broader audience according to this referee. Now, the current version addresses different aspects in the introduction and discussion sections, which have been significantly expanded along with the result section.

It is my suspicion that based on the seemingly rushed writing and awkward format that this was originally a Nature rejection that received a slapped together and boiler plate intro and discussion to conform to the Nat Com format. To be acceptable and relay the truly important and novel aspects of this work, the main text should contain: a concrete hypothesis, an explanation of the advancement in the single amplified virus genome approach (this is the best part), details of the reasoning for the experimental design, and a clear and complete explanation and discussion of the data.

As per suggestion of this referee, we have edited and structured the manuscript according to Nature Communications guidelines. This new version contains “a concrete hypothesis, an explanation of the advancement in the single amplified virus genome approach, details of the reasoning for the experimental design, and a clear and complete explanation and discussion of the data”. We hope that now this referee finds the new version more suitable.

In its current form this work is too narrow and misses an opportunity to really inform the community at large about the utility and limitations of this approach or what these viruses are or doing (other than being abundant).

In the new version of the manuscript, we present new results on proteomic data and expand the discussion section. The limitations of this approach is thoroughly addressed now in the last section of discussion.

Lastly, it is not clear how the increase in read lengths of these viral genomes contributes to our understanding of viral microdiversity. By my reading, it seems that what you found was in fact similar to other past works. Those networks reveal that your viruses are within the groups previously found but that some aspect (which I guess I missed) makes them more diverse at the nucleotide level. This seems to be the main novel finding yet it's not entirely fleshed out.

The results on microdiversity at the nucleotide identity and the biological implications of such discovery is now addressed in the updated result and discussion sections. We hope that this referee finds the new version more suitable for its publication and we appreciate the time for reviewing this manuscript.

Reviewer #2 (Remarks to the Author),

In “Single-virus genomics reveals hidden cosmopolitan and abundant viruses”, the authors developed a method to sequence single viral particles. They sorted and genomically amplified a total of 2,658 viral particles, for which 47 were randomly selected for genome sequencing. The aim of this study was to demonstrate that single cell viromics (SVG) is an unbiased way, as compared to metagenomics and culturing, to look at viral communities.

The manuscript is overall well written, but there are some sentences that need revisions and editing. A few examples:

Line 86: ... ”in spite of a handful...”

Line 123: ... ”putative hosts did not clarified...”

These two sentences have been modified in the new version of the manuscript.

In this study, they looked at single viral genomes from seawater and saliva samples. The saliva sample, as currently written in the manuscript, seems like an after thought and do not bring anything to the story, except confusion. There is only a short paragraph at the end discussing the single interesting virus that was found in saliva, with no figure related to it. It should either be removed completely, or more details should be added to discuss in depth the novelty of the presented method in the field of human viromics.

As two referees agree on the “saliva story” is out of focus, we have followed their recommendations and that part has been totally removed from the new version of the manuscript, as the “marine story” is much more solid and robust and stands by its own. To be honest, this issue was already discussed among authors, and we had a division of opinions, and finally we decided to let the last word to referees.

The microdiversity simulations are compelling and aim to answer a very important question in the field of environmental virology: why are assembled genomes in viromics not the most abundant viruses? This is a very important point that the authors make and should be emphasized. One thing that the authors did in their analysis, but did not present in the manuscript, is how biogeography impacts viral distribution and microdiversity. It would be interesting to choose a few of the most abundant viruses and show how their abundance changes depending on the virome used for metagenomic recruitment.

It has been included a new main figure (Fig. 3), supplementary figure (Supplementary Fig. 15) and detailed information in Results section on the biogeography for some of the most abundant viruses. However, we are working on a separate study that addresses in deep the microdiversity dynamics for some of these vSAGs discovered here and how that is linked to viral distribution by combining experiments in the field, digital PCR and in silico analyses of *Tara* viromes.

Fig. 3. Biogeography of most abundant marine viruses. The abundance of the most abundant surface dsDNA viruses for each virus genome datasets according to the procedure for genome recovering (single-virus genomics (red), viruses from single bacterial cells²⁷ (blue), virus cloned in fosmids (grey)¹¹, virus isolates (green) and viromics from *Tara* Oceans dataset^{2,3}. Fragment recruitment data was used to estimate the overall abundance for each region. Bubbles represent the fragment recruitment estimation expressed in KPKG (as in Fig. 2).

Supplementary Figure 15: Abundance distribution of the most abundant marine viruses. The abundance of the most abundant surface dsDNA viruses for each virus genome datasets according to the procedure for genome recovering (single-virus genomics (37-F6), viruses from single bacterial cells²⁷ (AAA164-I21), virus cloned in fosmids¹¹ (AP014248. putative Cyanophage), virus isolates (Pelagibacter phage HTVC010P) and viromics from *Tara* Oceans dataset^{2,3} (34DCM_32712), in all viromes. Fragment recruitment data was used to stimate the overall abundance for each region. Abundance is represented in KPKG (as in Fig. 2).

Single virus genomics has potential to look at viral diversity, as demonstrated in the manuscript, but also has disadvantages, which should be discussed, or at least pointed out in the manuscript. For example, no link to the host can be done, which limits ecological impact interpretations. On lines 122-123, the authors mention that they attempted to identify the host of the virus 37-F6 using tetramer frequency (Suppl. Fig. 8). This analysis is very weak and unnecessary. First, tetranucleotide frequency is not related to the host, and second, the other sequences consist only of a few related viruses. Obviously, closely related viruses will have more similar tetranucleotide frequencies. There are a few ways to link environmental viruses to their environmental hosts, which include looking at CRISPR inserts or tRNA genes.

We thank this referee for this comment. The new version now discusses in deep several disadvantages of single virus genomics, and in particular addresses the issue of missing “links” to the hosts in the final paragraph of the manuscript. In addition, more detailed information has been included regarding the CRISP-spacers, tRNA, as per suggestion of this referee.

Regarding the comment of this referee on tetranucleotide signatures, it has been widely used in different studies, for instance, recently in Roix et al (Nature, 2016). For a comprehensive literature on the use of in silico host prediction by k-mers (tri, tetramer frequency and so on), please refer to the Introduction section in the recent paper by Ahlgren et al., (2017 in Nucleic Acids Research; doi: 10.1093/nar/lgw1002) or in the review by Edwards et al., (2015; FEMS Microbiology).

On line 92, the authors again discuss the potential ecological impact of viruses by suggesting that they were actively infecting marine bacterioplankton. Single viral genomics in no way can determine if viruses are active. This sentence is pure speculation and should be either more detailed or removed.

Normally, when viral reads are found in a cellular metagenome, is because likely these viruses are actively replicating in a cell. In fact, that strategy of recovering viral sequences in a cellular metagenome, such as the *Tara* microbiome, has been widely used from other authors before to assemble the genome of active viruses infecting cells. For instance, in Mizuno et al., 2013 (PLoS Genetics), they described that approach and clearly stated:

-“it has been discovered that cellular metagenomes [...] contain significant amounts of viral DNA derived from cells undergoing the lytic cycle.”

-“We have sequenced and assembled ~6000 metagenomic fosmids obtained from the Mediterranean DCM (MedDCM) cellular fraction (>0.2 μ m). Among them more than a thousand genomic contigs were derived from marine phages that were actively replicating and are described here”.

Here, we have found reads from *Tara* microbiome that matched 95-100% identity with our viral genomes, and thus we consider as other authors have published and demonstrated, that these viral sequences are likely from virus actively replicating.

In any case, now, the sentence has been detailed (line 118) according to the suggestion of this referee.

Another disadvantage of SVG is that it is very expensive, therefore we can only look at a few viral genomes at a time, rather than the whole community as in viromics. The authors briefly mention that their dataset is limited on line 86, but didn't go further.

In Supplementary discussion section, we stated the following: “In the *Tara* virome survey³², with 5,476 viral contigs (109 Mb of assembled genome data and 2,16 billion raw reads) recruited up to 9.97%³². However, after normalization of recruitment rate according to total assembled genomic data, 1 Mb of single-virus genomic data would recruit \approx 3.5-fold more than data obtained by viromics (Supplementary Fig. 7). Finally, the overall sequencing effort carried out here to deliver 40 reference genomes compared to previous viromic surveys³² was significantly less, at least a 3-fold decrease”.

In any case, we discuss more in deep the advantages and disadvantages of SVG in the new version of the manuscript (line 279).

On line 64, the authors mention that the sample size necessary for SVG is very small (1 ml). Because of the small sample size, and the low number of genomes sequenced (47), the authors should be more cautious when discussing that they sequenced all the major viruses and viral groups.

We did not state in any part of the manuscript that we have “sequenced all the major viruses and viral groups”. What we state in general in our manuscript is that we have recovered some of the most uncultured dominant and abundant viruses in Surface Ocean from widespread and cosmopolitan clusters. In the last paragraph regarding to the comment of this referee, we state

“An additional benefit of SVGs is the very low sample volume requirement (typically ≤ 1 mL) to unveil the genomics of biologically relevant viruses”. **We fully agree with this referee that we have not sequenced all major viruses, but we would like to reiterate that this statement is not supported nor mentioned in any section of the manuscript.**

Reviewer #3 (Remarks to the Author),

Summary: The authors of this manuscript build on previous work where they had been able to successfully apply single cell genomics technology to virology. Here they take one step further and demonstrate these capabilities in environmental samples, a remarkable feat on its own. They were able to sequence 47 viral genomes (from marine and salivary samples) with different degrees of completeness; after a series of bioinformatics analysis they assigned taxonomy and clustered with previously known clusters of viral genomes. To place their findings in an ecologically relevant framework the authors then assessed the abundance of their viral genomes in different metagenomic and proteomic datasets, and the authors argue that their newly discovered viruses are among the most abundant in these global datasets. Interestingly, the authors note that one of these abundant virus 37-F6 had only loose similarity to a few viral-like contigs recovered from uncultured sources. These two contrasting observations are seemingly at odds: abundant in metagenomic samples yet barely ever seen, so the authors argue that it might be the high levels of intra-population variation that hinders assemblability of such abundant populations (I agree it is a good possibility, but a simulation would be more convincing). Finally the authors extend their proof of concept to another environment, human saliva. While there is a great deal of valuable information here and I applaud the approach in general as a significant advance in the field, I have several concerns about details of the analysis that are critical for the conclusions.

Regarding the comment of this referee on “a simulation would be more convincing”, that was exactly what we did, a simulation with real viromes and viral populations with different scenarios, low, medium and high levels of microdiversity and is described in detail in Methods, Results, Discussion and Supplementary Information sections. In the new version, according to this referee, the Results section has been expanded and now addresses and explains more in detail that specific part.

General criticisms:

The use of single cell technology had been previously adapted to virology, in the present case it is a remarkable improvement to be able to use in environmental samples. However, I am afraid that in the process of working to make their findings ecologically relevant, the authors apparently overlooked some important details in their bioinformatics analysis (specific details further down below):

1) Fragment recruitment; Nowhere in the methods did I see details of what kind of recruitment was used; to avoid difficulties I hope it was at least "competitive" (if not reciprocal best hit) and that it was done at the nucleotide level. The details are incredibly important to interpretation, and this lack of critical information makes evaluation difficult. Maybe it is fine. I suggest the authors should use the methods described in references 1,5 and 9.

We truly appreciate and thank this referee for the comment on the fragment recruitment methodology applied in this study because now the new version of the manuscript is likely even more solid.

We have done a step further and according to the suggestion of this referee, we have performed and applied for four representative viromes, the basic fragment recruitment methodology described in “Supplementary information” in ref. 1 (Zhao et al., 2013; pg. 11). We have applied the methodology described in that paper ref. 1: “Metagenomic sequences which returned a best-hit of the query genome from (2b) were identified and extracted from the metagenomic database” (pg. 11, step 3 in ref. 1 “Suppl Information”). So, each read was only assigned to one viral genome according to the best-hit scoring and finally, we calculated the relative recruitment as described. The results following that fragment recruitment methodology pointed to exactly same conclusions previously stated in the paper (line 228). A new supplementary figure (Supplementary Fig. 21) has been included in the manuscript. Thus, we are truly convinced that we have considered all possible variations of the methodology.

In addition, we would like to clarify, that it so happens that authors from references 5 and 9 cited by this referee several times are also authors in this manuscript. This referee specifically

discussed that references 5 and 9 (Mizuno et al., 2013; and Brum et al., 2015) used the same methodology as in ref. 1 (Zhao et al., 2013), and suggested that we indeed should use same or similar methodology as these references. However, as described in methodology of references 5 and 9 (see below), and given that authors of these two papers are also involved in this one, they did not apply the competitive fragment recruitment nor the reciprocal best hit approach in their surveys as clearly the method sections mention:

-Paper ref. 5, Brum et al 2015 (Science): *"The relative abundance of each population was computed by mapping all quality-controlled reads to the set of 5476 non redundant populations (considering only mapping quality scores greater than 1) with Bowtie 2"*

-Paper ref. 9 is Mizuno et al., (PLoS Genetics): *" For depicting comparative recruitment across metaviromes and metagenomes (as shown in Figure 5), a hit was considered if it was at least 50 bp long, had an e-value of less than 1e-5 and more than 95% [...]"*

In fact, the methodology applied for fragment recruitment in refs. 5 and 9 is different than that in ref 1. It is true, and we agree with this referee that best-hit recruitment and competitive fragment recruitment was used in ref. 1 (Zhao et al., 2013; "Pelahiphages's paper"). In addition, that approach has been used in a few other cases with bacterial genomes as well, such as in Santoro et al., 2014 (PNAS; for the description of "Candidatus Nitrosopelagicus brevis"). However, in these particular cases, as they wanted to discriminate the metagenomic fragment recruitment of very closely genetic related strains with very high average genome nucleic acid identity, they opted for the competitive version, which at the end, take into account only the recruitment of reads that map against specific genome regions or hypervariable regions of each one of these strains, avoiding core genome regions. In our case, as in refs. 5 and 9 mentioned by this referee, we do not have that scenario of very closely related phage strains. Thus, in this survey, we agreed to use the methodology described in ref. 5 and 9. In fact, our methods are very similar to ref. 9 (Mizuno et al., 2013). Nevertheless, according to this referee, we have followed her/his recommendation and performed the alternative analyses by using the methodology described in ref. 1 (Zhao et al 2013) to tests the impact on our conclusions, as explained above. This has been included in the new version and no changes nor major impacts have had on our initial conclusions. In any case, we thank this referee for this appreciation.

Finally, as we discussed in the manuscript, we also used other aligners, such as BWA and Bowtie as in ref. 5 besides BLAST program for one of the viromes, to perform our fragment recruitment results with the only aim to assess if the choice of the aligner software might impact on the fragment recruitment results. *"Finally, several programs are publically available to perform fragment recruitment, such as BLAST, BWA or Bowtie^{30,32,47,48}. We tested and compared recruitment results of vSAGs using BWA and Bowtie for the Mediterranean Tara Oceans metavirome dataset and no significant differences were obtained compared to BLAST and pointed to the same conclusions of the overwhelming relative recruitment rate of our vSAGs".*

We agree with this referee that more detailed information has to be given for the fragment recruitment methodology. Thus, the new version of the manuscript has been updated and we truly appreciate the comment of this referee to have an improved version of the manuscript. As we consider important this comment, we have included these new results following the referee's recommendation in the main text with a new Supplementary figure 21.

Supplementary Fig. 21. Comparison of different algorithms for metagenomic fragment recruitment. We compared the method that we used in our metagenomic fragment recruitment (Fig. 2) with the reciprocal-best hit fragment recruitment employed in other surveys (Zhao et al., 2013). Best-hit fragment recruitment was carried out with the Enveomics bioinformatic package (Rodriguez-R & Konstantinidis, 2016). Two fragment recruitment variants were also tested: without query coverage filtering and applying a 90% of query coverage cut-off. **a)** Fragment recruitment with three different viromes are showed, Benguela Current (BC066), Indian Monsoon (IM046), and Southern Atlantic (SA068), using a 70% and 95% Identity cut-off. **b)** Relative fragment recruitment with Benguela Current virome (BC066). **c)** Data of the three recruitments. Overall, data indicate that no differences were observed among recruiters.

2) Peptide recruitment; This one is very troubling, as proteomic peptides should not be aligned against a reference just with blast; Two fundamental things are different from nucleotide sequences: 1) These peptides are not random, as there is an enzymatic digestion that cleaves proteins at specific (and known) sites. 2) These are very short, often <10 AA residues long. These peptides are used to identify longer proteins based on multiple hits and non redundant matches; the latter is particularly important as peptides are short and will often be covering a domain that can be represented in multiple sequences. Additionally, when identity is lower, it can be matching only a few amino acid residues (potentially misleading, and databases are sparse compared to global diversity). As the authors appear very keen on using proteomic data, I would recommend they use the proteins themselves that were identified with these peptides in the original published work and not the peptides alone. Reference 28 identified proteins as structural, and such annotation was later propagated by the use of protein clusters within the Tara dataset. Also, some work used specialized spectral searchers such as Xtandem, Percolator or Sequest.

It so happens that authors from Brum et al (PNAS, 2015) cited by this referee are on board on this study, and we agree that the approach and methodology applied here seems suitable, and what is most important, results from peptide analyses obtained by mass spectrometry agree with other independent analyses, such as, metagenomic and viromic recruitment data. In any case, we have included in this new version more proofs as per request of referee 3 (see below for details, Fig. 5 and supplementary information of the paper)

In the *Journal of Proteomics*, which is a reference journal in that topic, there are several recent examples where authors undertook similar approaches as the one employed here to compare by BLASTp the resulting peptide sequences obtained by MS/MS against a protein database in order to identify the matches with previously described proteins; as in our case. For instance, in Martinez-Esteso et al., 2016, in Table 3A, they showed the BLASTp results of short peptides and used a very similar approach used here. Another example, is the paper by Colgrave et al., 2016 -three months ago- and we quoted “[...] peptides were selected and subjected to BLASTp analysis (NCBI BLASTp server) against all other cereals to ensure specificity to barley. For example, barley peptide marker G2 (VFLQQQCSPVR, B1-hordein, Uniprot: I6TMW0) was detected with high precursor intensity (> 10,000 counts) and yielded only 100% BLASTp matches against proteins from the taxonomy *Hordeum* [...]”. It is important to remark that in our research we only consider those hits that showed 100% query coverage and 100% identity, in other words, a perfect match. There are more examples on proteomics using similar strategy in other journals, such as Millares et al., 2012 (PLoS ONE, Tables S6 and S7). Moreover, the use of BLAST to compare short sequences against a database is widely used and accepted as a robust methodology, as long as the parameters of BLAST are adjusted for short sequences, as we did here according to BLAST manual. For instance, in the CRISPR-Cas field, which is a hot topic nowadays because of the potential use of cas9 protein for editing genomes, the comparison of spacer sequences, which are by nature very short in length, is conducted by BLAST (see and array of BLAST-based tools in the following link to compare short sequences from spacers or direct repeats; <http://crispr.i2bc.paris-saclay.fr/>). Currently, there are in the literature a vast number of papers using that strategy.

In viral proteomics (Brum et al. 2015), most obtained peptides are from structural proteins of the capsid, because basically, capsid proteins are the dominant proteins forming a virion. In our research, taking as an example the genome of virus 37-F6, which has over 20 ORFs, the results of BLASTp of peptides obtained from MS/MS by Brum et al (2015) validate our approach because the resulting matches (over 200 hits), are precisely with the predicted protein of the ORF no. 9, which is precisely a structural protein of the capsid (Fig. 2 and suppl information). Interestingly, we have predicted the 3D structure of that protein capsid and compared with that model previously published in Brum et al 2015. As referee could see, our 3D model is nearly identical to that proposed structure of the capsid protein for the cluster CAM_CRCL_773 in that PNAS paper by Brum and colleagues. In addition, the capsid protein of the virus 37-F6 that accumulated the highest number of predicted peptide sequences from MS/MS data, aligned perfectly with those capsid proteins from *Tara* expedition. Moreover, capsid protein of 37-F6 showed same conserved amino acid position in the alignment when compared to capsid proteins of CAM_CRCL_773 cluster. That new data is now a main figure of the manuscript (Fig. 5). Thus, we do think that, altogether, 3D-data modelling, amino acid alignment of capsid protein, BLASTp results agree and point to same idea, that what we are observing from the recruitment

data is indeed peptide sequences matching with actual structural protein. Thus, all data indicate that our BLASTp results are not the resulting bias from random BLASTp hits of peptides with non-structural proteins having similar domains. Abundance results from proteomics and metagenomics agree and point to the abundance of virus 37-F6, no matter how we look at the data, and results of BLASTp matches are not randomly distributed against different non-structural ORFs from virus 37-F6 because of the use of short sequences as queries for the analyses. If that would be the case, we would expect to obtain an array of random hits with non-structural proteins predicted from different ORFs along the genome for 37-F6. However, that is not the case, as we have shown, since peptides indeed matched only and specifically with structural capsid proteins. Thus BLASTp analyses are accurate enough and agree with those previous data published by Brum et al (PNAS).

In any case, taking the suggestion of this referee, we performed a similar searching approach described in that Brum et al paper. So, we took all predicted ORFs (protein sequences) from virus 37-F6 and were digested *in silico* by applying same parameters as in Brum et al paper (“fragment ion tolerance, 0.5; up to four missed cleavages allowed, variable modification of carboxymethyl cysteine (+57.021 Da) and fully tryptic peptides only”) with the bioinformatics tool PeptideMass implemented in EXPASY. http://web.expasy.org/peptide_mass/. Then the resulting theoretical digested peptides (Table 1; see below) were compared against the obtained peptide sequences from MS/MS data. We obtained a total of 209 matches with those Ms/MS data from peptides and nearly all matches were with digested peptides obtained from that structural capsid protein of virus 37-F6 previously identified and described in fig.2 and not with other proteins from the rest of predicted ORFs of 37-F6 genome. Thus, data point and lead to same conclusion and there are no significant differences among both methodologies. These result are now included in the new version of the manuscript (line 158) (see an example of the *in silico* digestion below). Therefore, we appreciate this referee for this comment on proteomic analyses.

Fig. 5. Capsid protein of vSAG 37-F6 and abundance in proteomic Tara viral dataset. (a) Peptide alignment of vSAG 37-F6 with the capsid proteins of cluster CAM_CRCL_773, by convenience we only show 8 sequences of the 152 total capsid proteins. Blocks represent the alignment of viral peptides from Tara expedition³¹ with the 37-F6 capsid protein, and

color denote the origin of peptides. Conserved amino acid position in the protein alignment are denoted with “*” (b) Representative 3D-structural model, using I-TASSER prediction server, of the 37-F6 capsid protein compared with the most similarity viral proteins, the *Tara* Contig 67SUR_4106 and SAGs AAA160-P02 and AAA164-I21. (c) Number of total recruited peptides from *Tara* expedition³¹ (100% identity and coverage cut offs) for the top two most recruiters viruses from each viral genomic dataset. Metaproteomics analysis show that gene 9 of vSAG 37-F6 encoded a high homology structural protein with the protein cluster CAM_CRCL_773, the most abundant viral marine protein.

Table: In silico digestion of the structural capsid protein of virus 37-F6

[Theoretical pI: 4.97 / Mw (average mass): 33228.26 / Mw (monoisotopic mass): 33207.41]

mass	position	#MC	artif.modification(s)	modifications	peptide sequence
1694.8150	137-184	1	MSO: 174, 180	1705.4783	TGDGVTLFNTAHPTVAGQFK NLTSTAADLNETSLEQSMID IAGMTDER
1599.0944	215-258	1	MSO: 228, 230, 257	1615.0893	TGTADNDINAIVSMGMVPQG YRVNNYLTDTDIFYIITDVP NGMK
1574.7421	60-101	1			AEGQGIFDEAQETFTARYT HETVALAFAITEEAIEDNLY DR
1274.2513	27-59	1	MSO: 49	1279.5830	YENQHAEIYTTENSDFRFE EVMLSGFANAQVK
1269.9317	43-77	1	MSO: 49	1275.2633	AFEEVMLSGFANAQVKAEG QGIFDEAQETFTAR
1199.9615	122-156	1			AVEPLINGLPNGSFKTGDGV TLFNTAHPTVAGQFK
1122.8766	157-187	1	MSO: 174, 180	1133.5398	NLTSTAADLNETSLEQSMID IAGMTDERGLR
1071.2017	78-105	1			YHETVALAFAITEEAIEDN LYDRLASR
1018.1563	237-262	1	MSO: 257, 259	1028.8195	VNNYLTDTDIFYIITDVPNG MKMFNR
1014.1410	157-184	0	MSO: 174, 180	1024.8043	NLTSTAADLNETSLEQSMID IAGMTDER
928.7836	78-101	0			YHETVALAFAITEEAIEDN LYDR
913.4377	211-236	1	MSO: 228, 230	924.1009	SQGRGTADNDINAIVSMGM VPQGYR
835.4053	237-258	0	MSO: 257	840.7369	VNNYLTDTDIFYIITDVPNG MK
781.7000	259-279	1	MSO: 259, 269	792.3633	MFNRAPLTTAMEGDFDTGNV R
770.6999	215-236	0	MSO: 228, 230	781.3632	TGTADNDINAIVSMGMVPQG YR
750.0789	6-25	1			SQLVKELEPGLNALFGLEYK
709.3226	26-42	1			RYENQHAEIYTTENSDFR
696.0018	263-281	1	MSO: 269	701.3335	APLTTAMEGDFDTGNVRYK
687.6848	137-156	0			TGDGVTLFNTAHPTVAGQFK
657.2889	27-42	0			YENQHAEIYTTENSDFR
652.9693	60-77	0			AEGQGIFDEAQETFTAR

637.6948	119-136	1			QVKA VEPLINGLPNGSFK
636.3425	195-210	1	MSO: 195, 209	647.0057	MIIPSELQFTAERLMK
623.9732	43-59	0	MSO: 49	629.3048	AFEEEEVMLSGFANAQVK
617.0000	11-26	1			ELEPGLNALFGLEYKR
606.9976	192-207	1	MSO: 195	612.3292	GVKMIIPSELQFTAER
598.9491	263-279	0	MSO: 269	604.2807	APLTTAMEGDFDTGNVR
576.6177	286-302	1			YSFGVSDPRGIFASPGA
564.9663	11-25	0			ELEPGLNALFGLEYK
519.2875	122-136	0			AVEPLINGLPNGSFK
512.2693	195-207	0	MSO: 195	517.6009	MIIPSELQFTAER
438.2142	284-294	1			ERYSGVSDPR
378.2217	1-10	1	MSO: 1	383.5534	MAIRSQVLK
350.1904	109-118	1	MSO: 114	355.5221	ALARSMSNAK
343.1663	286-294	0			YSFGVSDPR
331.5113	113-121	1	MSO: 114	336.8429	SMSNAKQVK
280.1642	102-108	1			LASRYTK
274.8326	106-112	1			YTKALAR
273.8217	208-214	1	MSO: 209	279.1533	LMKSQGR
248.1609	185-191	1			GLRVAAR
240.4623	295-302	0			GIFASPGA
234.1536	188-194	1			VAARGVK
213.1040	113-118	0	MSO: 114	218.4356	SMSNAK
193.1091	1-5	0	MSO: 1	198.4407	MAISR
192.1235	6-10	0			SQLVK
189.7618	259-262	0	MSO: 259	195.0934	MFNR
179.7763	280-283	1			YKAR
177.7714	282-285	1			ARER

Specific comments:

LINE 13: I believe the authors mean either 44, or 37, since it only becomes 47 after adding the salivary viruses.

Totally right and it has been modified in the abstract. The total number of marine vSAGs is 44. (line 14)

LINE 57: Most marine vSAGs... This sentence would benefit from including an N-number.

Data has been added to that sentence accordingly. "For most vSAGs (32 out of 44), a single large genome contig was obtained" (line 81)

LINE 97-101: The findings described here are exciting but I think that it might not be fully supported by the data; Please see below for further explanation on the metaproteomics analyses.

As explained above, new data in Result section with a new main figure is presented regarding the proteomic analyses. See above in “general criticisms” for the comprehensive answer to this concern

LINE 102-112: Readers will benefit from clarification on the kind of recruitment done by the authors (see general criticisms). Reference 1 includes a great description of the methods best suited to recruit metagenomic reads against a few novel references. The authors of 1 have a sensitivity analysis showing a decrease of recruitment as more closely related sequences are added. Notably the methods of 1 have been used in 5 and 9.

It seems to me that the best way to analyze this would be reciprocal best blast hit as in references 1,5 and 9. Perhaps that was done here, but it needs to be clarified.

This has been specifically addressed above in the answer for “General criticisms”. To avoid repetition and reiteration, please refer to that part where referee could find the answer to this concern.

LINE 146-154: It is unclear to me what the saliva viruses are adding to the story. Needs a justification beyond the fact it was done.

As two referees agree on the “saliva story”, we have followed their recommendations and that part has been totally removed from the new version of the manuscript. To be honest, this issue was already discussed among authors, and we have a clear division of opinions, and finally we decided to let the last word to referees.

LINE 182-206: The proteomics figure will need revision.

As discussed above in the “general criticism”, new data and figures have now presented according to this referee.

Methods Section:

LINE 468: How did the authors look for chimeric assembly (a significant potential problem)? Can you describe the methods and parameters used?

It has been included in the new version (line 612). We used Geneious R8 software in order to conduct a post-assembly to merge those contigs by using the criteria set in the program to only merge a contig if they showed 100% identity in the alignment with a minimum overlap of 200 bp with no gap in the overlapping. The overlapping/assembly of these contigs were manually checked one at a time to ensure that no mismatches and disagreements were present in the merged contigs as previously set by the program.

LINE 473: I assume the authors used a specific set of parameters with metavir? If so, please state them.

It has been modified in the new version (line 618). Parameters by default described in the Metavir paper ref. 47 and implemented in the online platform were used.

LINE 473-474: How did the authors predict Open Reading Frames? The methods skip directly to annotation without describing how genes were predicted. Please state the program and parameters used.

ORFs were predicted as described the Metavir’s paper (line 619). It has been modified in the new version accordingly. Metavir uses MetaGeneAnnotator.

LINE 474-474: The authors should describe exact setting used during their blast comparison to RefSeq Virus. Additionally, what kind of blast was used? Why did the authors use only the viral portion of RefSeq? Does that “stack the deck” or add bias?

This has been modified in the new version (line 618). Predicted ORFs from metavir platform were also compared in house by BLASTp against the non redundant (nr) Genbank database in order to corroborate and double check the annotation obtained through Metavir as an independent procedure. As we obtained nearly identical annotation, we considered that Metavir annotation was good enough and finally it was accepted as good and robust. We also used initially the virus refseq, but finally it did not provide any new information other than the already obtained from Metavir. Thus, it has been removed from the new version since is more reliable to use the whole nr database than the virus refseq one.

LINE 507: Can the authors please formalize “most prevalent” to an exact notation?

According to this referee, it has been modified in then new version (line 663) as follows: “Taxonomy predictions were based on the presence of reference sequences within each VC, with either 1) a “majority-rules approach” where the most abundant ($\geq 50\%$) reference sequence taxonomy being applied to all VC members (i.e. if 60% of reference sequences were Caudovirales, then the entire VC was classified as such) or 2) using a “lowest common ancestor” (LCA) approach among the reference sequences, where taxonomic lineages for each reference within the VC were compared to identify the lowest taxonomic rank (order, family, genus, etc) that contains all the reference sequences”

LINE 527: Can the authors describe why they only used the longest contigs during their recruitment work? Justification for excluding the shorter ones, and a particular cutoff?

For several viruses, we also obtained small contigs in the genome assembly, about 2kb or similar. However, we considered that is much informative to use the longest genome fragments that would represent better the recovered viruses for assessing their abundances. We consider that those short fragments do not provide meaningful information other than the data already provided by the recruitment taking the longest fragment that truly represent better the viral genome. In some viromic papers, they use an artificial cut-off, such as 10 kb (e.g. Brum et al paper, ref 5), for estimating abundances. That is a pragmatic cut-off without any special scientific or technical reason supporting that value, other than a practical decision.

LINE 530: As noted above, it is not clear if the recruitment work was done competitively or not, i.e. is each read only mapped once? Since this is a central finding, arguments about abundance and prevalence are based on results from it, the authors are recommended to use the methods in Reference 1, 5 and 9. Whatever they used, the authors should describe it more clearly. If not competitive or reciprocal best hit, it could be problematic.

To avoid reiteration, please refer to our answer for the “general criticism (point 1)”, where we address all general and specific concerns regarding to this issue.

LINE 534: Reference 60 is not appropriate, in that work uses TEM data and not any kind of -omics; Perhaps the authors meant reference 5, with the same lead author?

Yes, we meant reference 5. It has been modified in the new version (line 693)

LINE 539: I believe Tara Oceans Expedition, not Malaspina.

Modified in the new version (line 698).

LINE 538-543: Proteomics analyses should not be done with the use of the peptide recovered from the referenced work. There are fundamental differences from metagenomic reads: 1) Peptides are short 2) They are not random.

Regarding the second point, it is not clear if the authors ensured that the area matching these peptides within their genomes was forced to be flanked in the C-terminal by a Lysine or Arginine (for example from data from reference 28 where trypsin was used in the digestion).

These peptides are recovered by matching different spectra against a reference database; when multiple peptides can be identified we can only be sure a protein was identified if a protein has both multiple spectra associated to it and if it has unique non redundant spectra associated to it. The latter point is especially important as peptides could often, due to their short nature, cover only a part of a domain that repeats in multiple proteins. Not a small issue.

If the authors wish to use proteomic data, two alternatives are recommended:

1) To use the proteins that the authors of 28 and 29 identified from these spectra (or peptides) and blast them as one normally would in the case of protein sequences.

2) To search all the spectra with specialized searchers (xtandem, sequest, percolator) including their own genomic sequences in the database + plus whatever it was used by the works 28 and 29.

To avoid reiteration, please refer to our answer for the “general criticism (point 2)”, where we address all general and specific concerns regarding to this proteomic issue.

LINE 544 and after:

Methods do not include details on the construction of the phylogenies and they are not referenced within the main text. The authors should note details on the construction, program, algorithm, etc.

We consider that is not necessary to give the details of methods because we employed same methodology as in reference 9 and 18 (Mizuno et al., 2013 and 2016) of the previous version of this manuscript. In fact, that is properly cited and mentioned in the method section.

Authors of these two studies are also in this single virus paper and we all considered redundant to give again all details when they are already published.

LINE 552: The authors should provide details on the kind of QC that the reads had before going into alignment for SNP calling. QC is described for reads coming from their own work (a few sections before) but not for reads taken from Tara, for example. Such details can matter and should be included.

We totally agree with this referee and more detailed information about QC filtering for Tara have been included (line 710). As referee could see in this new version (method section), only reads passing the QC filtering from Tara have been used for the analyses. In fact the QC filtering used in that *Tara*'s survey and our study with the Blanes virome were almost identical since reads were removed when the median quality score was <20 and bases were trimmed at the 3' end of reads if the quality score was <20. Thus, only reads with high quality score were considered for the SNP analyses to avoid the potential impact of reads with bad quality on SNPs analyses.

REVIEWERS' COMMENTS:

Reviewer #1 (Remarks to the Author):

This is a dramatically improved revision of the original work. The addition of additional text on the background topic and discussion of the findings has placed the paper within a broader context that will be more appealing to the Nature Communications community.

At the same time the authors have addressed a majority of my issues including the nice new addition of a figure that better visualizes the unique insight that vSAGs can generate. Lastly the more elaborate discussion of the how this method was conducted much improves the manuscript.

Reviewer #2 (Remarks to the Author):

In "Single-virus genomics reveals hidden cosmopolitan and abundant viruses", the authors developed a method to sequence single viral particles. They sorted and genomically amplified a total of 2,234 viral particles from marine samples (epi-, meso-, and bathypelagic), for which 44 were randomly selected for genome sequencing. The aim of this study was to demonstrate that single cell viromics (SVG) is an unbiased way, as compared to metagenomics and culturing, to look at viral communities.

This new version of the manuscript is much improved from the previous version. The details added to the manuscript make the reading easier to understand, limit incorrect interpretation of the data, and make it a more enjoyable read. The authors now state a clear hypothesis regarding the fact that metagenomics assemblies are hindered by high intra-population viral diversity. The hypothesis is then tested using microdiversity analyses. The removal of the saliva samples also helped refocusing the analysis to discuss only the main findings of single cell viromics.

Minor comments:

Not sure why single cell genomics and viromics' acronyms are SCGs and SGVs, respectively. It would make more sense to write them as in other papers, SCG and SCV.

The number of sorted particles is different in the abstract and on line 75.

Supplementary figure 12: vSAGs are labeled as SAVGs.

Reviewer #3 (Remarks to the Author):

Response to general comments 1. As the authors well know, fragment recruitment of metagenomic datasets comes in many flavors; I commend the authors for expanding their

methodology to use multiple algorithms and furthermore summarizing them on a concise well-presented figure. I have no additional comments in this regard as their final approach now its convincing. Finally I think that their summary of recruitment algorithms can be of value to the community.

Response to general comments 2. I thank the authors for expanding their metaproteomics search to include theoretically generated peptides from your metagenomes. This is particularly important, as proteomic peptides are not random. Showing that these match the peptides found by Brum et al. adds another, more convincing, line of evidence to your findings.

Additionally, clarifying the matching of these peptides occurs only at 100%ID /100% Coverage improves the veracity of their findings.

As above I am glad these new observations were summarized on a new main figure that offers a direct link indicating the ecological relevance of their study. I have no additional comments on this regard.

Specific comments (Lines referred to originally received document)

LINE 13: Fully addressed now.

LINE 57: No additional comments.

LINE 97-101: No additional comments.

LINE 102-112: Fully addressed elsewhere, no more comments

LINE 146-154: No additional comments.

LINE 182-206: No additional comments, I am happy to see a new more comprehensive figure.

Methods Section:

LINE 468: Thank you for expanding these methods, no additional comments.

LINE 473: No additional comments.

LINE 473-474: No additional comments.

LINE 474-474: Fully addressed now.

LINE 507: No additional comments

LINE 527: No additional comments.

LINE 530: Addressed elsewhere, no more comments.

LINE 534: No more comments

LINE 539: No more comments

LINE 538-543: Addressed elsewhere, no more comments.

LINE 544 and after: No additional comments.

LINE 552: No additional comments.

REVIEWERS' COMMENTS:

Reviewer #1 (Remarks to the Author):

We only would like to thank this referee for the valuable comments on this manuscript. As it is stated below all concerns have been fully addressed

This is a dramatically improved revision of the original work. The addition of additional text on the background topic and discussion of the findings has placed the paper within a broader context that will be more appealing to the Nature Communications community.

At the same time the authors have addressed a majority of my issues including the nice new addition of a figure that better visualizes the unique insight that vSAGs can generate. Lastly the more elaborate discussion of the how this method was conducted much improves the manuscript.

Reviewer #2 (Remarks to the Author):

We agree with this referee that the new version, after editing the manuscript according to this referee's comments, is more enjoyable, and we thank the time on reviewing this manuscript. As it is stated below all concerns have been fully addressed

In “Single-virus genomics reveals hidden cosmopolitan and abundant viruses”, the authors developed a method to sequence single viral particles. They sorted and genomically amplified a total of 2,234 viral particles from marine samples (epi-, meso-, and bathypelagic), for which 44 were randomly selected for genome sequencing. The aim of this study was to demonstrate that single cell viromics (SVG) is an unbiased way, as compared to metagenomics and culturing, to look at viral communities.

This new version of the manuscript is much improved from the previous version. The details added to the manuscript make the reading easier to understand, limit incorrect interpretation of the data, and make it a more enjoyable read. The authors now state a clear hypothesis regarding the fact that metagenomics assemblies are hindered by high intra-population viral diversity. The hypothesis is then tested using microdiversity analyses. The removal of the saliva samples also helped refocusing the analysis to discuss only the main findings of single cell viromics.

Minor comments:

Not sure why single cell genomics and viromics' acronyms are SCGs and SGVs, respectively. It would make more sense to write them as in other papers, SCG and SCV.

Acronym of SCGs comes from Single Cell Genomics (initial letters). Thus, here using same reasoning, the “correct acronym” for Single Virus Genomics be the initial letters, SVGs.

The number of sorted particles is different in the abstract and on line 75.

It has been modified accordingly.

Supplementary figure 12: vSAGs are labeled as SAvGs.

It has been modified accordingly to vSAGs

Reviewer #3 (Remarks to the Author):

We are very glad that now this reviewer finds more convincing our proteomic and metagenomic fragment recruitment results. As it is stated below, all concerns have been fully addressed. Her/his comments have been very stimulating to have a better version of our manuscript.

Response to general comments 1. As the authors well know, fragment recruitment of metagenomic datasets comes in many flavors; I commend the authors for expanding their methodology to use multiple algorithms and furthermore summarizing them on a concise well-presented figure. I have no additional comments in this regard as their final approach now its convincing. Finally I think that their summary of recruitment algorithms can be of value to the community.

Response to general comments 2. I thank the authors for expanding their metaproteomics search to include theoretically generated peptides from your metagenomes. This is particularly important, as proteomic peptides are not random. Showing that these match the peptides found by Brum et al. adds another, more convincing, line of evidence to your findings.

Additionally, clarifying the matching of these peptides occurs only at 100%ID /100% Coverage improves the veracity of their findings.

As above I am glad these new observations were summarized on a new main figure that offers a direct link indicating the ecological relevance of their study. I have no additional comments on this regard.

Specific comments (Lines referred to originally received document)

LINE 13: Fully addressed now.

LINE 57: No additional comments.

LINE 97-101: No additional comments.

LINE 102-112: Fully addressed elsewhere, no more comments

LINE 146-154: No additional comments.

LINE 182-206: No additional comments, I am happy to see a new more comprehensive figure.

Methods Section:

LINE 468: Thank you for expanding these methods, no additional comments.

LINE 473: No additional comments.

LINE 473-474: No additional comments.

LINE 474-474: Fully addressed now.

LINE 507: No additional comments

LINE 527: No additional comments.

LINE 530: Addressed elsewhere, no more comments.

LINE 534: No more comments

LINE 539: No more comments

LINE 538-543: Addressed elsewhere, no more comments.

LINE 544 and after: No additional comments.

LINE 552: No additional comments.